# SLAM/SAP signaling regulates discrete γδ T cell developmental checkpoints and shapes the innate-like γδ TCR repertoire

Somen K Mistri[1], Brianna M Hilton[1], Katherine J Horrigan[1], Emma S Andretta[1], Remi Savard[1], Oliver Dienz[1], Kenneth J Hampel[2], Diana L Gerrard[2], Joshua T Rose[2], Nikoletta Sidiropoulos[2], Dev Majumdar[1], Jonathan E Boyson[1]*

[1]Department of Surgery, Larner College of Medicine, University of Vermont, Burlington, United States; [2]Department of Pathology and Laboratory Medicine, Larner College of Medicine, University of Vermont Medical Center, Burlington, United States

*For correspondence:
jboyson@uvm.edu

Competing interest: The authors declare that no competing interests exist.

## eLife assessment

This **important** study highlights the importance of SLAM-SAP signaling in determining innate gamma-delta T cell sublineages and their T cell receptor repertoires. It uncovers the complex role of the SLAM-SAP pathway in developing specific gamma-delta T cell subsets. The evidence presented is **compelling**, backed by high-quality data obtained through advanced single cell proteogenomics techniques. This work will be of broad interest to immunologists.

**Abstract** During thymic development, most γδ T cells acquire innate-like characteristics that are critical for their function in tumor surveillance, infectious disease, and tissue repair. The mechanisms, however, that regulate γδ T cell developmental programming remain unclear. Recently, we demonstrated that the SLAM/SAP signaling pathway regulates the development and function of multiple innate-like γδ T cell subsets. Here, we used a single-cell proteogenomics approach to identify SAP-dependent developmental checkpoints and to define the SAP-dependent γδ TCR repertoire in mice. SAP deficiency resulted in both a significant loss of an immature $Gzma^+Blk^+Etv5^+Tox2^+$ γδT17 precursor population and a significant increase in $Cd4^+Cd8^+Rorc^+Ptcra^+Rag1^+$ thymic γδ T cells. SAP-dependent diversion of embryonic day 17 thymic γδ T cell clonotypes into the αβ T cell developmental pathway was associated with a decreased frequency of mature clonotypes in neonatal thymus, and an altered γδ TCR repertoire in the periphery. Finally, we identify TRGV4/TRAV13-4(DV7)-expressing T cells as a novel, SAP-dependent Vγ4 γδT1 subset. Together, the data support a model in which SAP-dependent γδ/αβ T cell lineage commitment regulates γδ T cell developmental programming and shapes the γδ TCR repertoire.

## Introduction

Like αβ T cells, γδ T cells can be broadly grouped into innate-like IFN-γ-producing γδT1, IL-4-producing γδT2, or IL-17-producing γδT17 subsets that are functionally programmed during thymic development (*Grigoriadou et al., 2003*; *Haas et al., 2012*; *Ribot et al., 2009*; *Sumaria et al., 2017*), or that undergo differentiation in the periphery (*Chien et al., 2013*). These innate-like γδ T cells are highly enriched in mucosal tissues such as the skin, gut, and lung and play critical roles in the host response to pathogens, tumor surveillance, autoimmunity, and tissue homeostasis. Accumulating data suggest that the relative balance of these functional γδ T cell subsets can have important implications in

disease susceptibility and progression. This is perhaps best exemplified in the setting of cancer where IFN-γ-producing γδT1 have been shown to play a critical role in tumor elimination (*He et al., 2010*; *Gao et al., 2003*; *Girardi et al., 2001*), while γδT17 subsets promote tumor growth and metastasis (*Coffelt et al., 2015*; *Wu et al., 2014*; *Rei et al., 2014*; *Wakita et al., 2010*). The thymic developmental programs, therefore, that regulate the balance of these functional subsets in the periphery may have considerable impact on disease outcome. Unlike the case for their αβ T cell counterparts, however, the mechanisms that determine whether a developing γδ T cell acquires an innate-like γδT1 or γδT17 phenotype during thymic development remain unclear.

A significant body of data supports a model in which weak or strong TCR signal strength might contribute to the development of innate-like γδT17 (weak signal) and γδT1 (strong signal) subsets (*Sumaria et al., 2017*; *Turchinovich and Hayday, 2011*; *Jensen et al., 2008*; *Muñoz-Ruiz et al., 2016*; *Chen et al., 2021*), and a strong TCR signal has been suggested to be necessary for the development of the IFN-γ- and IL-4-producing γδNKT cell subset (*Kreslavsky et al., 2009*). However, there is also evidence of a TCR-independent model in which pre-existing hard-wired transcriptional programs regulate the development of some innate-like γδT17 subsets (*Spidale et al., 2018*), and accumulating data suggest that γδT17 programming is associated with the use of distinct signaling pathways (*Laird et al., 2003*; *Sumaria et al., 2021*; *Muro et al., 2018*).

SLAM family receptors comprise a family of nine receptors, SLAMF1 (SLAM; CD150), SLAMF2 (CD48), SLAMF3 (Ly9; CD229), SLAMF4 (2B4; CD244), SLAMF5 (CD84), SLAMF6 (Ly108; CD352), SLAMF7 (CRACC; CD319), SLAMF8 (BLAME), and SLAMF9 (SF2001; CD84H), that are expressed primarily on hematopoietic cells and which play diverse roles in immune development and function (*Cannons et al., 2011*). Most SLAM family receptors interact in a homophilic manner, the exception to this rule being SLAMF4, which serves as the ligand for SLAMF2. SLAM family receptor signals can be both activating or inhibitory, depending on the recruitment of the cytosolic signaling adapter proteins SAP (encoded by *Sh2d1a*) and EAT-2, and inhibitory SHP-1, SHP-2, or SHIP phosphatases to immunoreceptor tyrosine switch motifs in the SLAM family receptor cytoplasmic domain (*Cannons et al., 2011*). Although SLAM/SAP signaling has long been known to play a role in innate-like NKT (*Nichols et al., 2005*) and γδNKT (*Kreslavsky et al., 2008*) cell development, recent findings from multiple groups demonstrate that this pathway is also involved in the development of innate-like γδT1, γδT17 (*Dienz et al., 2020*) and innate-like MAIT cells (*Legoux et al., 2019*; *Koay et al., 2019*). Together, these findings indicate that SLAM/SAP signaling plays a fundamental role in the development of a majority of the innate-like T cell lineages.

In this study, we set out to define the stage of thymic development during which SAP mediates its effect on γδ T cell developmental programming and to define SAP-dependent changes in the γδ T cells transcriptional program during their development. Finally, since SAP-dependent innate-like T cell subsets are often characterized by highly restricted TCRs (e.g., Vγ1⁺/Vδ6.3⁺ γδNKT; *Azuara et al., 1997*), we wished to identify whether specific TCR clonotypes were also associated with SAP-dependent γδT1 and γδT17 T cells. Therefore, we utilized a single-cell proteogenomics approach with V(D)J profiling that allowed us to simultaneously compare cell surface protein expression, gene expression, and TCR clonotypes between B6 and B6.*Sh2d1a⁻/⁻* γδ T cells from embryonic, neonatal, and adult thymus, as well as from peripheral lung γδ T cells. Collectively, these data provide a comprehensive map of the SAP-dependent γδ T cell subsets and their developmental checkpoints in perinatal and adult thymus.

## Results

### SLAM family receptor expression marks transcriptionally distinct developmental stages among E17 γδ T cells

γδT17 developmental programming is thought to occur primarily during embryonic/neonatal thymic development (*Haas et al., 2012*; *Spidale et al., 2018*; *Havran and Allison, 1988*), and is tightly linked to the expression of discrete TCR gamma and delta pairings (*O'Brien and Born, 2010*). To identify the specific developmental stages of thymic development during which the SLAM/SAP signaling pathway exerts its effects, we employed a single-cell RNAseq approach coupled with feature barcoding (i.e., CITEseq; *Stoeckius et al., 2017*) and V(D)J profiling to compare embryonic day 17 (E17) B6 and B6.*Sh2d1a⁻/⁻* thymic γδ T cells (*Figure 1A*). (Fluorescence activated cell sorting) FACS-sorted thymic γδ

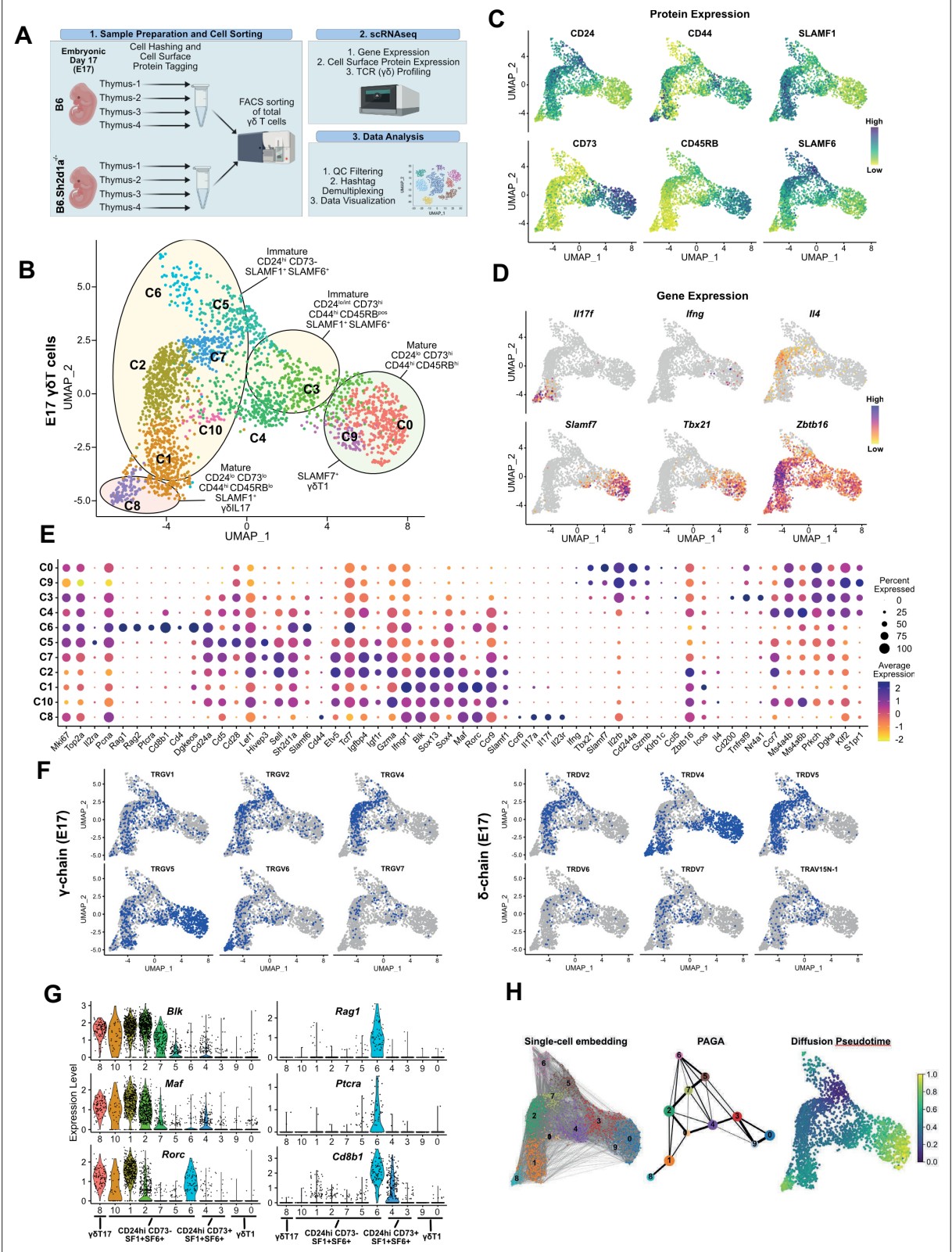

**Figure 1.** SLAM family receptor expression marks transcriptionally distinct developmental stages among E17 γδ T cells. (**A**) Schematic workflow depicting the methodology for single-cell RNA sequencing (scRNAseq) library preparation and subsequent data analysis pipeline employed in this study. (**B**) Uniform manifold approximation and projection (UMAP) visualization displaying 11 distinct clusters of E17 B6 thymic γδ T cells, *n* = 4 individual mice. Clusters are annotated based on comprehensive protein and gene expression data. (**C**) Feature plots illustrating the cell surface protein

*Figure 1 continued on next page*

*Figure 1 continued*

expression profiles of CD24, CD73, CD44, CD45RB, SLAMF1, and SLAMF6 on B6 E17 thymic γδ T cells. Each data point represents a cell, color-coded to indicate varying protein expression levels (high: dark blue, low: yellow). (**D**) Feature plot illustrating the gene expression profiles of signature genes among individual B6 E17 thymic γδ T cells. Each data point represents a cell, color-coded based on gene expression levels (high: purple, low: yellow). (**E**) Dot plot demonstrating the scaled expression levels of selected genes in E17 B6 thymic γδ T cells. Normalized expression levels are depicted using a color scale ranging from low (yellow) to high (purple). Dot size corresponds to the fraction of cells within each cluster expressing the specific marker. (**F**) UMAP representation of E17 B6 thymic γδ T cells indicating the expression of selected TRGV (TCRγ; left) and TRDV (TCRδ; right) chain V-segment usage (in blue) across individual cells. (**G**) Violin plots illustrating the expression patterns of selected genes among E17 B6 thymic γδ T cell clusters. (**H**) Visualization of single-cell trajectories using PAGA (partition-based graph abstraction) with single-cell embedding (left) showing connectivity between individual nodes (middle). Weighted edges represent statistical measures of interconnectivity. The diffusion pseudotime plot (right) delineates inferred pseudotime progression of cells along developmental trajectories using cluster 5 (**C5**) as the root, highlighting their developmental order (from purple to yellow).

The online version of this article includes the following figure supplement(s) for figure 1:

**Figure supplement 1.** Quality control for single-cell CITE-seq.

**Figure supplement 2.** TCR repertoire profiling of B6 E17 thymus γδ T cells.

**Figure supplement 3.** Identification of a BLK^neg^MAF^neg^RORγt^pos^ E17 γδ T cell population expressing CD4 and CD8.

T cells (*Figure 1—figure supplement 1*) from individual embryos (*n* = 4 B6 and 4 B6.*Sh2d1a^-/-^* mice) were identified using hashtags, and Vγ1+ and Vγ4+ γδ T cells were identified using barcoded antibodies (ADTs). To help define the developmental trajectories of the E17 γδ T cells, we also stained cells with barcoded antibodies specific for CD24, CD73, CD44, and CD45RB, which define distinct developmental pathways of γδT17 and γδT1 cells in the thymus (*Sumaria et al., 2017*; *Coffey et al., 2014*; *In et al., 2017*), as well as for SLAMF1 and SLAMF6. After data filtering and QC, we verified that clusters were evenly distributed among the biological replicates (*Figure 1—figure supplement 1*). The resulting dataset represented a total of 5278 thymic γδ T cells from B6 (*n* = 2597) and B6.*Sh2d1a^-/-^* (*n* = 2681) γδ T cells. Data were visualized using uniform manifold approximation and projection (UMAP) for dimensionality reduction and a graph-based clustering approach (*Satija et al., 2015*).

First, we annotated the B6 dataset using a combination of cell surface protein expression, gene expression, and TCR clonotype sequence. This analysis revealed 11 B6 E17 γδ T cell clusters, mostly distributed between two major branches (*Figure 1B*). We observed the presence of CD24^low^73^low^44^high^45RB^low^SLAMF1^pos^ (c8) and CD24^low^73^high^44^high^45RB^high^SLAMF6^low/pos^ (c0, c9) clusters, consistent with mature γδT17 and γδT1 phenotypes, respectively (*Figure 1C*, *Figure 1—figure supplement 1*). In support of this annotation, the c8 cluster was enriched in *Il17a*, *Il17f*, *Il23r*, *Rorc*, and TRGV6/TRDV4 (IMGT nomenclature; encoding Vγ6/Vδ1) transcripts with limited CDR3 diversity, while the c0/c9 cluster was enriched in *Ifng*, *Tbx21*, *Klrd1*, *Il2rb*, *Xcl1*, and TRGV5/TRDV4 (encoding Vγ5/Vδ1) transcripts also with little to no diversity (*Figure 1D–F*, *Figure 1—figure supplement 2*, *Supplementary file 1 and 2*, *Havran et al., 1991*; *Itohara et al., 1990*). Consistent with previous observations (*Dienz et al., 2020*), we noted that the c0/c9 cluster was highly enriched in *Cd244a* (*Slamf4*), and was enriched in transcripts from another SLAM family member, *Slamf7* (*Figure 1D, E*). Flow cytometric analysis confirmed that SLAMF7 was expressed primarily on mature γδ T cells and that its expression was mostly confined to γδ T cells expressing the canonical CD44+CD45RB+ markers of γδT1 cells (*Figure 1—figure supplement 1*, *Sumaria et al., 2017*; *Wencker et al., 2014*).

We identified a small CD24^low/int^73^high^44^high^45RB^pos^ c3 cluster characterized by enrichment in *Cd200* (whose expression is upregulated upon TCR ligation; *Buus et al., 2017*), *Il2rb*, *Nr4a1*, *Cd244a*, *Xcl1*, and *Tnfrsf9* (encoding 4-1BB). A high frequency of invariant TRGV5/TRDV4 clonotypes and decreased TCR clonotype diversity (*Figure 1—figure supplement 2*, *Supplementary file 2*) suggested that c3 represented an immature γδT1 population. The CD24^low/pos^CD73^low/pos^44^low/pos^CD45RB^low/pos^ c4 cluster was enriched in *Ms4a4b and Ms4a6b* (encoding CD20 homologues that modulate T cell activation; *Xu et al., 2010*; *Howie et al., 2009*), *S1pr1*, *Il6ra*, *Cd8a*, *Cd8b1*, and *Ccr9*, suggesting that the c4 cluster represents the previously described emigrating thymic *Ccr9+S1pr1+* γδ T cells (*Sagar et al., 2020*, *Figure 1C, E*). Consistent with the c4 cluster representing naive emigrating thymic γδ T cells, the diversity of the c4 TCR clonotypes was the highest among the CD24^low/int^ (c0/c9, c8, c3, c4) clusters (*Figure 1—figure supplement 2*).

We identified several immature CD24^high^CD73^neg^ ADT clusters (c1, c2, c7, c5, c6, and c10) that also expressed SLAMF1 and/or SLAMF6 (*Figure 1B*, *Figure 1—figure supplement 1*). The c1 and c2

clusters were enriched in γδT17-associated genes such as *Blk*, *Maf*, *Sox13*, *Ccr2*, *Tcf12*, and *Rorc* (*In et al., 2017*; *Laird et al., 2010*; *Zuberbuehler et al., 2019*; *Narayan et al., 2012*; *Lu et al., 2015*), as well as *Sh2d1a* and *Ccr9* (*Chen et al., 2021*, *Figure 1E*, *Supplementary file 1*). Within these two clusters, the CD44$^{low/pos}$CD45RB$^{low}$ c2 cluster was especially enriched in genes previously implicated in γδ T cell development such as *Etv5*, *Tox*, *Sox4*, *Tcf7*, *Zbtb16* (*Lu et al., 2015*; *Jojic et al., 2013*; *Malhotra et al., 2013*; *He et al., 2022*; *Fahl et al., 2021*), *Igfbp4*, and *Igf1r* which were recently implicated in Th17 differentiation (*DiToro et al., 2020*), *Tnfsf11* (RANKL) which regulates Aire$^+$ mTEC development (*Roberts et al., 2012*), as well as *Themis* and *Tox2* (*Figure 1E*, *Figure 1—figure supplement 1*). A high level of *Gzma* expression identified c2 as the previously described *Gzma*$^+$ γδ T cell subset (*Sagar et al., 2020*). Interestingly, in addition to expressing γδT17-associated genes, the c2 cluster was also enriched in γδT2-associated *Gata3* and *Il4* transcripts (*Figure 1D, E*, *Supplementary file 1*). The CD44$^{low/pos}$CD45RB$^{neg}$ c1 cluster, in contrast, exhibited significantly higher levels of *Il17a*, *Il17f*, *Rorc*, *Maf* (*Zuberbuehler et al., 2019*), as well as *Ifngr1*, consistent with the c1 cluster representing immature γδT17 cells (*Figure 1D–E, G*, *Figure 1—figure supplement 1*).

Analysis of the c1 and c2 TCR transcripts revealed that the c2 cluster was notably enriched in TRGV4-expressing cells and that 87% of Vγ4 T cells utilized just 3 TRDV chains (50% TRDV5, 24% TRDV2-2, and 12% TRAV13-4/DV7) (*Figure 1F*, *Supplementary file 2*). Clonotype analysis revealed a high frequency of semi-invariant TRGV4/TRDV5 and TRGV4/TRDV2-2 clonotypes previously associated with γδT17 (*Wei et al., 2015*; *Kashani, 2015*; *Sim and Augustin, 1990*, *Figure 1—figure supplement 2*, *Supplementary file 2*). In contrast, the c1 cluster possessed comparatively fewer of these TRGV4 clonotypes, and was enriched in TRGV6, TRGV5, and TRGV1 chains paired with TRDV4 (*Figure 1—figure supplement 2*, *Supplementary file 2*). Indeed, we noted that the Vγ1 repertoire, which is diverse in adult tissues, was dominated by TRGV1/TRDV4 clonotypes (35% of all TRGV1) in E17 thymus, and just two closely related semi-invariant TRGV1/TRDV4 clonotypes comprised 17.4% of all TRGV1. The location of these clonotypes in the immature γδT17-related clusters (*Figure 1—figure supplement 2*) suggested that they represented the dominant IL-17-producing Vγ1 clonotypes at E17, which we verified using flow cytometry (*Figure 1—figure supplement 3*).

Since the IL-4-producing Vγ1/Vδ6.3 γδNKT cell subset develops in fetal thymus (*Grigoriadou et al., 2003*), we asked whether the *Il4*-enriched γδ T cells in cluster 2 (*Figure 1D*) were immature Vγ1 γδNKT cells with previously identified canonical Vδ6.3 sequences (*Azuara et al., 1997*). Interestingly, while we did identify a small number of Vγ1 T cells with these canonical sequences (encoded by TRAV15-1/DV6-1 or TRAV15N-1) in our dataset, these were not present in the *Il4*-enriched c2 cluster.

The CD44$^{pos}$CD45RB$^{pos}$ c5 cluster was enriched in *Il2ra*, *Scin*, *Bex6*, *Cd5*, *Slamf6*, *Cd28*, *Hivep3*, and cell cycle/metabolism genes, consistent with γδ T cells that have recently developed from DN2/DN3 (*Sagar et al., 2020*; *Scaramuzzino et al., 2022*; *Prinz et al., 2006*). Within the c5 and c7 clusters, we noted that while there was expression of γδT17-associated genes such as *Blk*, *Maf*, *Sox13*, and *Etv5*, there was relatively little *Rorc* expression. *Rorc* expression became apparent in the c2 cluster (*Rorc*$^{low}$) and was significantly increased in the c1 cluster (*Figure 1—figure supplement 1*). Flow cytometric analysis confirmed BLK and PLZF expression in CD25$^{high}$ E17 Vγ4 T cells, while RORγt expression was primarily observed in the CD25$^{neg}$ population, suggesting that BLK is expressed soon after γδ T cell selection, followed by RORγt (*Figure 1—figure supplement 3*). These data were consistent with MAF-directed expression of *Rorc* (*Sagar et al., 2020*; *Zuberbuehler et al., 2019*) and suggested a c5–c7–c2–c10/c1–c8 γδT17 developmental trajectory, which was supported by trajectory inference analysis (*Figure 1H*).

Finally, analysis of the CD24$^{high}$73$^{neg}$44$^{neg}$45RB$^{int}$SLAMF1$^{pos}$SLAMF6$^{pos}$ c6 population revealed that it was highly enriched in *Cd8a*, *Cd8b1*, *Cd4*, *Rag1*, *Rag2*, *Ptcra*, *Arpp21*, and *Dgkeos*, consistent with cells developing along the αβ T cell lineage (*Figure 1E, G*, *Figure 1—figure supplement 1*, *Supplementary file 1*, *Mingueneau et al., 2013*). Analysis of the Vγ1 and Vγ4 ADTs confirmed that c6 cells were not simply contaminating αβ T cells as both Vγ1- and Vγ4-expressing cells were present in the c6 cluster (*Figure 1—figure supplement 3*) and analysis of the c6 TCR clonotypes revealed the presence of TRGV1, GV4, GV5, GV6, and GV7 clonotypes, none of which appeared to be specific to the c6 cluster (*Supplementary file 2*).

Since trajectory inference analysis suggested a strong relationship between the *Il2ra*$^+$ c5 and c6 clusters (*Figure 1H*), we conducted differential gene expression analysis between the c5 and c6 from B6 mice. Interestingly, we found that the c6 cluster was highly enriched in *Rorc*, suggestive of a γδT17

phenotype, but was mostly lacking in the expression of *Maf*, *Blk*, *Ccr2*, and other canonical γδT17-associated transcripts (*Figure 1E, G*, *Figure 1—figure supplement 1*). Indeed, flow cytometric analysis of E17 TCRβ$^{neg}$TCRδ$^{pos}$ γδ T cells confirmed that CD4CD8 DP γδ T cells exhibited an immature CD44$^{low}$CD25$^{low}$MAF$^{neg}$BLK$^{neg}$RORγt$^{pos}$ phenotype (*Figure 1—figure supplement 3*), and that RORγt expression in the absence of BLK or MAF is a characteristic of Lin$^{neg}$ DN thymocytes as they progress from CD44$^{low}$25$^{high}$ DN3 to CD44$^{low}$25$^{low}$ DN4 (*Figure 1—figure supplement 3*). Together, these data suggested that the c6 cluster γδ T cells represented previously described CD4CD8 DP γδT cells that were developing along an alternative αβ T cell pathway (*Passoni et al., 1997*; *Kang et al., 1998*; *Haks et al., 2005*).

## SAP regulates γδ T cell clonotype diversion into both the γδT17 and the αβ T cell developmental pathways

To identify SAP-dependent developmental checkpoints, we next compared the hashtag-separated B6 and B6.*Sh2d1a*$^{-/-}$ E17 γδ T cell datasets. A comparison of the cluster frequencies between B6 and B6.*Sh2d1a*$^{-/-}$ γδ T cells revealed significantly decreased frequencies of the CD24$^{high}$73$^{neg}$44$^{int}$45RB$^{int}$-SLAMF1$^{pos}$SLAMF6$^{pos}$ c2 and c1 clusters in B6.*Sh2d1a*$^{-/-}$ mice, and a striking increase in the frequency of the CD24$^{high}$73$^{neg}$44$^{neg}$45RB$^{neg}$SLAMF1$^{high}$SLAMF6$^{high}$ c6 cluster (*Figure 2A*). A comparison of differentially expressed genes among the immature (c1, c2, c7, c5, c6, and c10) clusters using pseudobulk analysis revealed 234 differentially expressed genes with a p$_{adj}$ threshold <0.005 and log$_2$ fold change greater than 0.5, with 148 genes exhibiting decreased expression in B6.*Sh2d1a*$^{-/-}$ γδ T cells, and 86 genes exhibiting increased expression (*Figure 2B*, *Supplementary file 3*). Consistent with the decreased frequencies of the c1/c2 clusters in B6.*Sh2d1a*$^{-/-}$ mice, a significant fraction of the differentially expressed genes that were downregulated in B6.*Sh2d1a*$^{-/-}$ mice reflected γδT17-associated genes (e.g., *Maf*, *Blk*, *Ccr2*, *Sox13*, *Etv5*, etc.) that were present in the c2 and c1 clusters, as well as a number of c1/c2-enriched genes (e.g., *Cpa3*, *Tox2*, *Pdcd1*, *Tmem121*, *Bex6*, and *Scin*) whose importance in γδ T cell development is unclear (*Figure 2B, C*, *Supplementary file 2*). This was not solely due to the decreased frequency of the c2 and c1 clusters in B6.*Sh2d1a*$^{-/-}$ mice, as a closer analysis revealed a decreased number of transcripts for many signature genes (e.g., *Blk*, *Etv5*, *Ccr2*) prevalent in the c2 cluster and to a lesser extent the c7 cluster (*Figure 2—figure supplement 1*). Given the decreased expression of *Blk*, we also examined the expression of Src tyrosine kinase family members *Lck* and *Fyn*, which have previously been implicated in T cell development (*Laird and Hayes, 2010*). Unlike *Blk*, which was mostly restricted to γδT17 clusters, *Lck* and *Fyn* expression was more broadly distributed (*Figure 1—figure supplement 1*). Using pseudobulk analysis to compare their expression between B6 and B6.*Sh2d1a*$^{-/-}$ immature E.17 γδ T cells, we found that neither *Lck* nor *Fyn* expression was appreciably altered (*Figure 2—figure supplement 1*), suggesting that *Blk* expression was uniquely dependent on SAP. We did note, however, that the magnitude of *Lck* differential expression was close to the 0.5 log$_2$ FC cut-off (*Figure 2—figure supplement 1*). Indeed, a large number of genes whose expression was increased in B6.*Sh2d1a*$^{-/-}$ mice were those whose expression was mostly confined to the c6 cluster (e.g., *Cd4*, *Cd8b1*, *Ptcra*, *Rag2*) or whose expression was highest in c6 (e.g., *Tcf7*, *Thy1*, *Lck*) (*Figure 2B, C*, *Figure 2—figure supplement 1*). Together, these data identified the *Gzma*$^+$*Etv5*$^+$*Sox13*$^+$*Blk*$^+$*Tox2*$^+$ c2 and *Cd4*$^+$*Cd8b1*$^+$*Ptcra*$^+$*Rag2*$^+$ c6 clusters as SAP-dependent developmental checkpoints.

Next, we asked whether these SAP-dependent alterations in γδ T cell development were associated with any changes in the TCR repertoire. A comparison of all γδ TCR clonotypes between E17 B6 and B6.*Sh2d1a*$^{-/-}$ γδ T cells revealed no significant changes in overall clonotype frequency (*Figure 2D*). However, a closer examination of the TCR repertoire revealed significant changes in the distribution of specific TCR clonotypes. For example, within the TRGV4 repertoire, which represents the dominant TCR among immature clusters at E17, we observed a significant decrease in TRGV4/TRDV5 clonotypes in the c2 cluster and a concomitant increase of these clonotypes in the c6 cluster (*Figure 2E*). Among the TRGV4 clonotypes, this appeared to be specific to the TRDV5 clonotypes, as we observed no significant differences c2 and c6 frequencies among TRGV4/TRDV2-2 and TRDV7 clonotypes (*Figure 2E*). We did note, however, a significant increase in the TRGV4/TRDV7 clonotypes in the *Il2ra*$^+$ c5 cluster (*Figure 2E*). Within the TRGV1 repertoire, we noted a significant increase in TRGV1/TRDV4 clonotypes within the c6 cluster (*Figure 2E*). Together, these data suggested that SAP operated at an early stage of γδ T cell development during or soon after the down-regulation of *Il2ra*, and that it

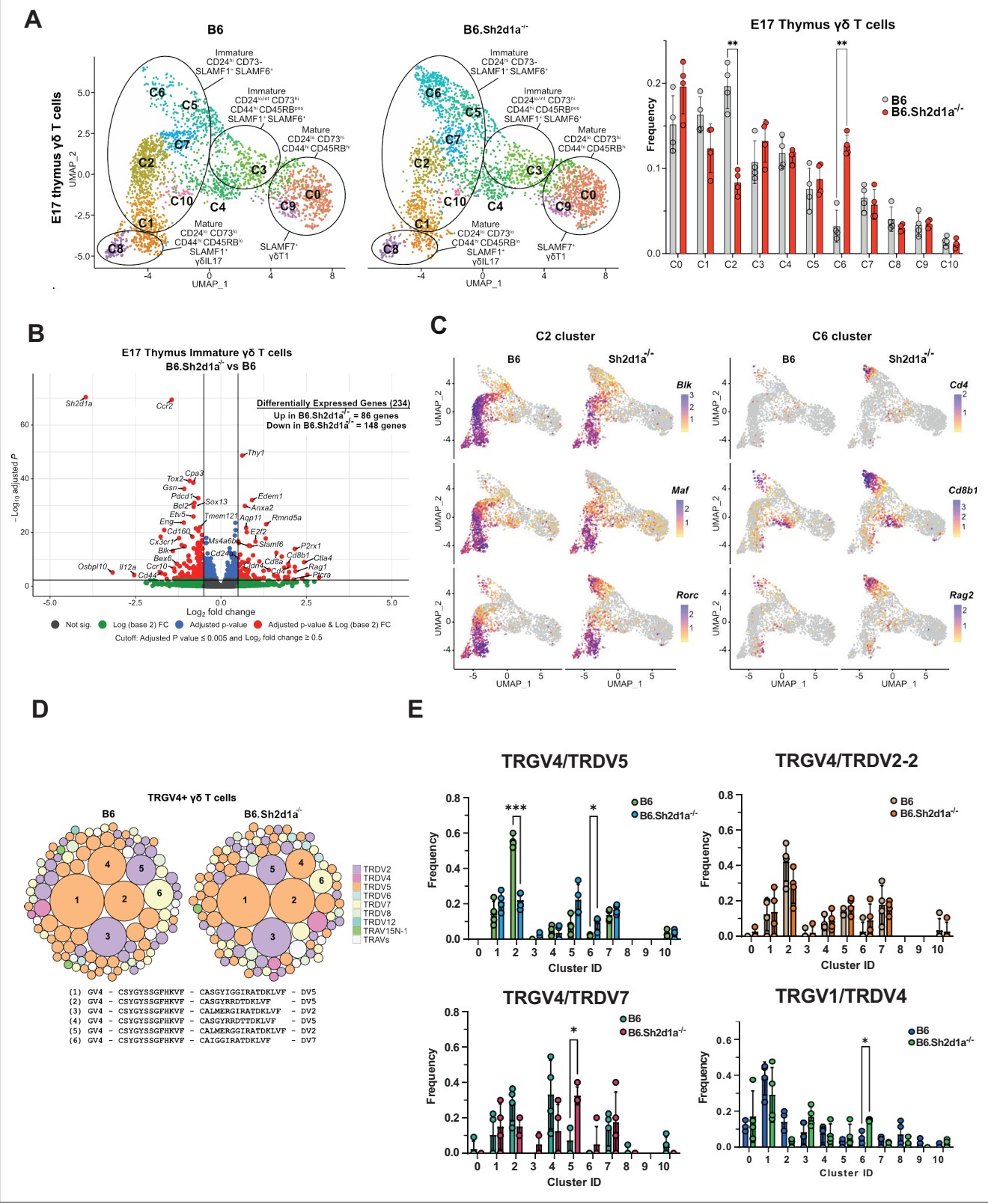

**Figure 2.** Identification of SAP-dependent developmental checkpoints during E17 γδ T cell developmental programming. (**A**) Uniform manifold approximation and projection (UMAP) representation of B6 (left) and B6.*Sh2d1a*⁻/⁻ (right) γδ T cells from E17 thymi (*n* = 4 mice per group) is shown at *left*. Clusters were annotated based on protein and gene expression data. The frequencies of B6 and B6.*Sh2d1a*⁻/⁻ E17 γδ T cell clusters are shown at right. Bars represent the mean cluster frequency, error bars represent standard deviation, **p ≤ 0.01, two-way ANOVA, Sidak multiple comparisons test;

*Figure 2 continued on next page*

*Figure 2 continued*

$n$ = 4 mice/genotype. (**B**) Volcano plot of differentially expressed genes among immature CD24$^{high}$CD73$^{neg}$ (clusters: C1, C2, C5, C6, C7, and C10) B6 and B6.*Sh2d1a*$^{-/-}$ γδ T cells using pseudobulk scRNAseq analysis. Genes exhibiting a log$_2$FC ≥0.5 or ≤ –0.5, and a p$_{adj}$ ≤ 0.005 in are shown in red. Only some selected genes are labeled for the sake of clarity. (**C**) Feature plots illustrating gene expression profiles of selected C2 and C6 cluster-specific genes among individual B6 and B6.*Sh2d1a*$^{-/-}$ E17 thymic γδ T cells. Each point represents a cell, color-coded by gene expression level (high: purple, low: yellow). (**D**) TCR clonotype bubble plot displaying the top 100 B6 and B6.*Sh2d1a*$^{-/-}$ TRGV4 clonotypes in E17 thymus. Bubbles represent unique TRGV4/TRDV clonotypes; size indicates frequency of clonotype among all Vγ4 T cells ($n$ = 388 B6, 288 B6.*Sh2d1a*$^{-/-}$ cells), and colors denote specific TRDV chains utilized. Selected clonotypes are numbered and their respective CDR3γ and CDR3δ sequences displayed below. (**E**) Altered TRGV/TRDV clonotype distribution between B6 and B6.*Sh2d1a*$^{-/-}$ E17 thymic γδ T cells. Frequencies of selected clonotypes among different clusters of B6 and B6.*Sh2d1a*$^{-/-}$ E17 γδ T cells is shown. Bars represent the mean clonotype frequency among the indicated TRGV/TRDV pairings shown, error bars represent standard deviation, *p ≤ 0.05, ***p ≤ 0.001, two-way ANOVA, Sidak multiple comparisons test; $n$ = 4 mice/genotype-age groups.

The online version of this article includes the following source data and figure supplement(s) for figure 2:

**Source data 1.** TCR clonotype distribution among E.17 B6 and B6.*Sh2d1a*$^{-/-}$ gamma delta T cells.

**Figure supplement 1.** Expression levels of selected genes among individual B6 and B6.*Sh2d1a*$^{-/-}$ E17 thymic γδ T cell clusters.

positively regulated entry into the γδT17 pathway at the c5/c7 to c2 transition, while at the same time inhibiting entry into the CD4$^+$CD8$^+$ αβ T cell-like c6 cluster. The net result of these changes appeared to be that specific TCR clonotypes such as TRGV4/TRDV5 were diverted from the γδT17 pathway into the αβ T cell-like pathway.

## SAP deficiency is associated with increased numbers of immature CD4$^+$CD8$^+$RORγt$^+$ thymic γδ T cells

Next, we independently evaluated these observations using flow cytometry. This analysis revealed that while SAP deficiency did not affect the overall number of E17 γδ T cells (*Figure 3—figure supplement 1*), it did result in a significant decrease of immature CD24$^{pos}$BLK$^{pos}$MAF$^{pos}$RORγt$^{pos}$γδ T cells in E17 B6.*Sh2d1a*$^{-/-}$ thymus (*Figure 3A, B*). We found that this decrease was most pronounced in the CD44$^{pos}$ population after downregulation of CD25 (*Figure 3A*, *Figure 3—figure supplement 1*) and that this population exhibited a high level of PLZF expression, suggesting it corresponded to the CD44$^+$*Blk*$^+$*Maf*$^+$*Zbtb16*$^+$*Rorc*$^+$ c1 cluster (*Figure 3—figure supplement 1*). In addition, UMAP analysis of E17 Vγ4 T cells flow cytometric data revealed the presence of a SAP-dependent CD24$^{pos}$BLK$^{pos}$PLZF$^{pos}$RORγt$^{neg}$ population in B6.*Sh2d1a*$^{-/-}$ thymus (*Figure 3—figure supplement 1*) consistent with the phenotype of the *Rorc*$^{low}$ c2 cluster. In contrast, we found no evidence of a SAP-dependent decrease in the CD25$^{pos}$ population frequency, consistent with the notion that SAP exerts its effect during or after this stage of development (*Figure 3A*). We also noted that in addition to its effect on the numbers of immature thymic γδT17, SAP deficiency resulted in the reduced expression levels of some of these markers, most notably BLK (*Figure 3C*), a finding that was consistent with the decreased number of *Blk* transcripts observed in *Figure 2*. Interestingly, as PLZF induction is often associated with SLAM/SAP signaling, we noted that SAP did not affect overall PLZF expression levels in E17 thymic γδ T cells (*Figure 3—figure supplement 1*).

In contrast to the decreased numbers of CD24$^{pos}$BLK$^{pos}$MAF$^{pos}$PLZF$^{pos}$RORγt$^{pos}$ γδ T cells, we observed a significant increase in the frequency and number of CD24$^{pos}$BLK$^{neg}$MAF$^{neg}$PLZF$^{neg}$RORγt$^{pos}$ γδ T cells (*Figure 3A, B*), whose phenotype was consistent with the αβ T cell-like c6 cluster in *Figure 2*. Consistent with our finding that the c6 cluster was enriched in CD4 and CD8, we found that the frequency and number of CD4 and CD8-expressing CD24$^{pos}$ TCRδ$^{pos}$TCRβ$^{neg}$ BLK$^{neg}$MAF$^{neg}$RORγt$^{pos}$ thymic γδ T cells was significantly increased in B6.*Sh2d1a*$^{-/-}$ mice (*Figure 3D*). Together, these analyses confirmed our single-cell CITEseq analysis, and suggested that SAP regulated the throughput of immature γδ T cells in both the γδT17 cell pathway and the αβ T cell-like pathway. More specifically, the data suggested that SAP promoted progression to the CD24$^{pos}$ BLK$^{pos}$MAF$^{pos}$PLZF$^{pos}$RORγt$^{low}$ c2 cluster that is enriched in a number of genes (e.g., *Sox13*, *Etv5*, *Blk*, *Maf*) known to regulate γδT17 development, but that it inhibited entry into the CD24$^{pos}$44$^{neg}$BLK$^{neg}$MAF$^{neg}$RORγt$^{pos}$ αβ T cell-like pathway (c6 cluster).

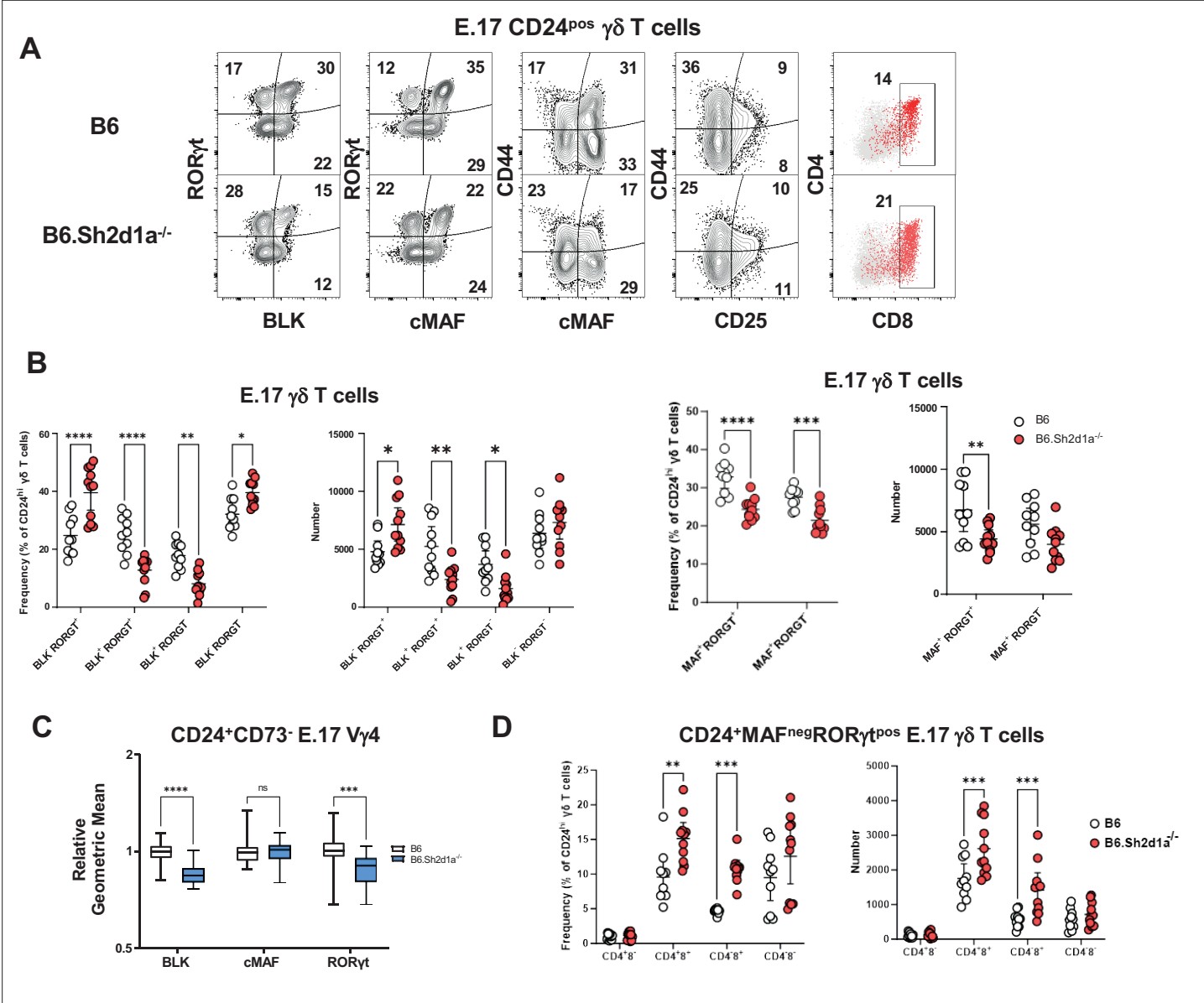

**Figure 3.** Increased frequency and number of immature CD4+CD8+c-MAF-RORγt+ γδ T cells in B6.*Sh2d1a*-/- E17 thymus. (**A**) Representative contour plots depicting CD24pos E17 γδ T cells in B6 (top) and B6.*Sh2d1a*-/- (bottom) mice. Concatenated data from two B6 and three B6.*Sh2d1a*-/- embryos are shown, and are representative of two independent experiments. Numbers in the plots represent the frequency as a percentage of CD24pos γδ T cells. (**B**) Cumulative frequency and number of immature CD24pos BLK- (left), c-MAF- (right), RORγt-expressing E17 thymic γδ T cells. The mean and standard deviation are indicated. Data are the cumulative data from two independent experiments, *n* = 10 B6, 11 B6.*Sh2d1a*-/- pooled embryonic thymi, *p ≤ 0.05, **p ≤ 0.01, ***p ≤ 0.001, ****p ≤ 0.0001, two-way ANOVA, Sidak's multiple comparisons test. (**C**) Relative geometric mean of BLK, c-MAF, and RORγt expression in B6 and B6.*Sh2d1a*-/- E17 CD24+CD73- Vγ4 γδ T cells, *p ≤ 0.001, ****p ≤ 0.0001, two-way ANOVA, Sidak's multiple comparisons test. Data are representative of two independent experiments, 7–9 mice per group. (**D**) Cumulative frequency (left) and number (right) of immature CD4- and CD8- MAFnegRORγtpos E17 thymic γδ T cells in B6 and B6.*Sh2d1a*-/- thymus. The mean and standard deviation are indicated. Data are the cumulative data from two independent experiments, *n* = 10 B6, 11 B6.*Sh2d1a*-/- pooled embryonic thymi, **p ≤ 0.01, ***p ≤ 0.001, ****p ≤ 0.0001, two-way ANOVA, Sidak's multiple comparisons test.

The online version of this article includes the following source data and figure supplement(s) for figure 3:

**Source data 1.** Frequency and number of immature CD4+CD8+ cMaf-RORγt+ E17 γδ T cells.

**Figure supplement 1.** SAP-dependent regulation of immature BLK+PLZF+RORγt+ and BLK-PLZF+/-RORγt+ E17 thymic γδ T cells.

**Figure supplement 1—source data 1.** Frequency and number of uniform manifold approximation and projection (UMAP)/FlowSOM clustered E17 thymic Vγ4 T cells.

# Identification of SAP-dependent developmental checkpoints during neonatal and adult thymus γδ T cell developmental programming

The neonatal period is a time of rapid expansion of numerous γδ T cell subsets in the thymus including Vγ4 γδT17 and γδNKT. Therefore, we conducted a comparison between B6 and B6.*Sh2d1a*$^{-/-}$ neonatal and adult thymic γδ T cells, to identify novel SAP-dependent developmental checkpoints and to track the outcome of the SAP-dependent alterations in γδ T cell development observed in E17 thymus. Neonate datasets revealed a similar branched clustering pattern as in E17 with some notable differences. Similar to E17, we observed immature CD24$^{pos}$73$^{low}$SLAMF1$^{pos}$SLAMF6$^{pos}$ clusters (c0, c1, c2, c3, c9, c12, c14), CD24$^{pos}$73$^{low/pos}$SLAMF1$^{pos}$SLAMF6$^{pos}$ (c4, c5, c10, c15) *S1pr1*-enriched 'naive' clusters, mature CD24$^{neg}$73$^{low}$SLAMF1$^{pos}$SLAMF6$^{neg}$ (c6, c7) γδT17 clusters, and a mature CD24$^{neg}$73$^{pos}$SLAMF1$^{neg}$SLAMF6$^{low/pos}$*Slamf7*$^{high}$ (c8) γδT1 cluster (**Figure 4A, B**, **Figure 4—figure supplement 1**). In general, we noted good correlation with cluster-specific gene expression between the E17 and D9 (day 9) datasets. A notable exception was a striking decrease in *Zbtb16* expression among all immature neonatal γδ T cells, which is consistent with previous reports (**Chen et al., 2021**; **Sagar et al., 2020**, **Figure 4—figure supplement 1**, **Supplementary file 3**).

In contrast to the E17 dataset, the neonatal thymic γδ T cell dataset contained a CD24$^{neg}$73$^{pos}$SLAMF1$^{neg}$SLAMF6$^{high}$*Slamf7*$^{neg}$ (c13) γδT2 cluster enriched in *Il4*, *Il13*, *Zbtb16*, *Cd40lg*, and *Nos1* (**Figure 4—figure supplement 1**, **Supplementary file 1**), suggesting it represented IL-4-producing γδNKT cells. Indeed, examination of the γδT2 (c13) and γδT1 (c8) cluster TCR repertoires revealed a significant enrichment of canonical TRGV1 and TRAV15N-1 or TRAV15-1/DV6-1 sequences characteristic of γδNKT cells (**Figures 4C and 5**; **Azuara et al., 1997**). Analysis of the differentially expressed genes in the γδT1 (c8) and γδT2 (c13) clusters revealed that *Zbtb16*, *Icos*, and *Slamf6* were markers of γδT2 cells while *Tbx21*, *Klrb1c* (NK1.1), and *Slamf7* were markers of γδT1 cells in neonatal thymus (**Figure 4—figure supplement 1**). Flow cytometric analysis confirmed that PLZF$^{hi}$ γδT2 Vγ1Vδ6.3 cells co-express SLAMF6 and ICOS while Tbet$^{hi}$ γδT1 Vγ1Vδ6.3 cells are NK1.1$^+$ and are either SLAMF6$^-$SLAMF7$^+$ or SLAMF6$^+$SLAMF7$^+$ (**Figure 4—figure supplement 2**).

A comparison of B6 and B6.*Sh2d1a*$^{-/-}$ neonatal γδ T cells revealed multiple SAP-dependent changes in clustering. Among the immature γδ T cells, we observed a decreased frequency of the immature CD24$^{high}$73$^{neg}$SLAMF1$^{pos}$SLAMF6$^{pos}$ c1 cluster (**Figure 4D**) that was enriched in *Blk*, *Etv5*, *Sox13*, *Rorc*, and *Maf* (**Figure 4—figure supplement 1**), suggesting it was analogous to the SAP-dependent c2 cluster in E17 γδ T cells. Similar to our observations in E17 γδ T cells, flow cytometric analysis confirmed both a decrease in BLK$^{pos}$RORγt$^{pos}$ and BLK$^{pos}$RORγt$^{neg}$ γδ T cell populations, as well as a decreased level of BLK expression (**Figure 4E, F**). In addition, we observed an increased frequency of immature CD24$^{high}$73$^{neg}$SLAMF1$^{pos}$SLAMF6$^{pos}$ cluster with characteristics of αβ T cells (c14) in B6.*Sh2d1a*$^{-/-}$ D9 thymic γδ T cells (**Figure 4D**, **Figure 4—figure supplement 1**). While this increase did not reach the level of significance, presumably due to the very low number of c14 cells in the dataset, flow cytometric analysis revealed a significant increase in immature CD24$^{pos}$73$^{neg}$BLK$^{neg}$RORγt$^{pos}$ γδ T cells, a phenotype consistent with γδ T cells that have been diverted to the αβ developmental pathway (**Figure 4E**). Last, we observed an increased frequency of immature CD24$^{pos}$73$^{low/pos}$ SLAMF1$^{pos}$SLAMF6$^{pos}$ *S1pr1*-enriched c4 and c5 clusters, but not the closely related *Cd200*-enriched c10 cluster (**Figure 4D**). While we did confirm an increased number of CD200$^{neg}$S1pr1$^{pos}$ γδ T cells in neonatal thymus by flow cytometry, this increase was relatively small and its significance is unclear (**Figure 4—figure supplement 2**). Independent confirmation of these SAP-dependent changes in neonatal γδ T cell gene expression was obtained by conducting bulk RNAseq analysis of FACS-sorted CD24$^{high}$ neonatal thymic γδ T cells from B6 and B6.*Sh2d1a*$^{-/-}$ mice (**Supplementary file 4**).

Among the mature neonatal γδ T cells, we observed a decreased frequency of the CD24$^{low}$73$^{neg}$SLAMF1$^{pos}$F6$^{low}$ γδT17 c6 cluster, which was consistent with our previous observation of decreased IL-17 production in B6.*Sh2d1a*$^{-/-}$ neonatal γδ T cells (**Dienz et al., 2020**). Interestingly, we observed a striking loss of the CD24$^{low}$73$^{high}$SLAMF1$^{low}$F6$^{high}$F7$^{neg}$ γδT2 c13 cluster in B6.*Sh2d1a*$^{-/-}$ mice, as well as a decreased frequency of the mature CD24$^{low}$73$^{high}$SLAMF1$^{low}$F6$^{low}$F7$^{high}$ γδT1 c8 cluster (**Figure 4A–D**). These findings were consistent with an SAP-dependent loss of both the IL-4- and IFN-γ-producing γδNKT and non-γδNKT γδT1 cells (discussed further below). Unfortunately, while we were able to define mature SAP-dependent γδNKT subsets, we found no evidence of immature

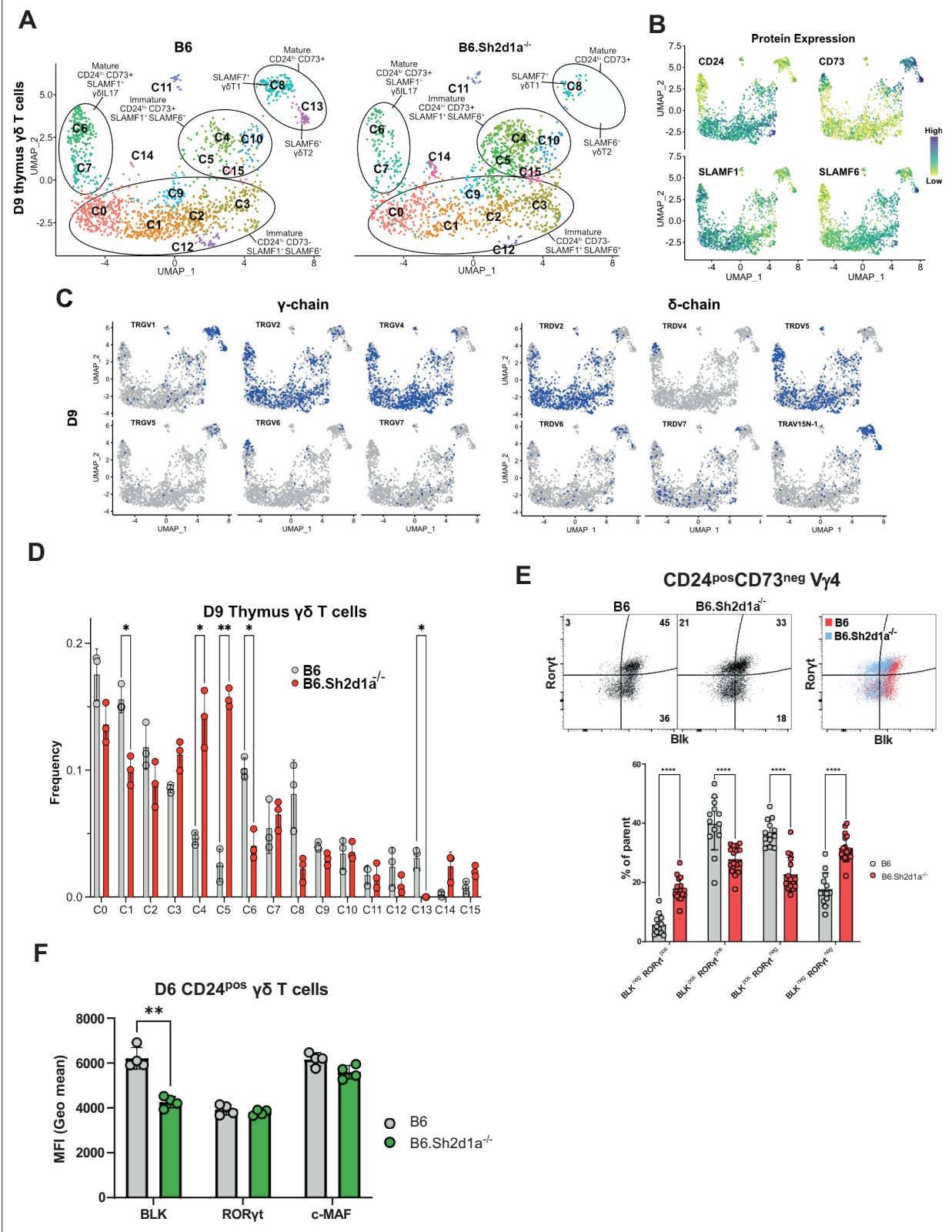

**Figure 4.** Identification of SAP-dependent developmental checkpoints during neonatal γδ T cell developmental programming. (**A**) Uniform manifold approximation and projection (UMAP) representation of B6 (left) and B6.*Sh2d1a*-/- (right) γδ T cells from D9 neonatal thymi (*n* = 3 mice per strain). Cluster annotation is based on comprehensive protein and gene expression data. (**B**) Feature plots displaying cell surface protein expression patterns of CD24, CD73, SLAMF1, and SLAMF6 on D9 B6 thymus γδ T cells. Data are color-coded based on protein expression level (high: dark blue, low: yellow). (**C**)

*Figure 4 continued on next page*

*Figure 4 continued*

UMAP representation of D9 B6 thymic γδ T cells exhibiting selected TRGV4 (left) and TRDV (right) chain V-segment usage (in blue) in individual cells. (**D**) Frequencies of B6 and B6.*Sh2d1a*^-/- D9 thymic γδ T cell clusters. Bars represent the mean cluster frequency, error bars represent standard deviation, *p ≤ 0.05, **p ≤ 0.01, two-way ANOVA, Sidak multiple comparisons test, 3 mice/strain. (**E**) Representative contour plots depicting BLK and RORγt expression in B6 and B6.*Sh2d1a*^-/- CD24^pos CD73^neg D9 Vγ4 T cells are shown *above*. Cumulative frequencies of BLK and RORγt expression are shown *below*. The mean and standard deviation are indicated. Data are the cumulative data from three independent experiments, n = 13 B6, 16 B6.*Sh2d1a*^-/- mice, ****p ≤ 0.0001, two-way ANOVA, Sidak's multiple comparisons test. (**F**) Decreased BLK expression in immature B6.*Sh2d1a*^-/- γδ T cells. Data represent the mean expression of the indicated proteins in D6 neonate BLK^pos, RORγt^pos, or cMaf^pos thymic γδ T cells, **p ≤ 0.01.

The online version of this article includes the following source data and figure supplement(s) for figure 4:

**Source data 1.** Comparison of BLK, RORγt, and cMaf expression among neonatal thymic γδ T cells.

**Figure supplement 1.** Transcriptional heterogeneity in neonatal thymic γδ T cells.

**Figure supplement 2.** Flow cytometric analysis of D9 thymic γδ T cells.

**Figure supplement 2—source data 1.** Phenotypic characterization of neonatal thymic γδ T cells.

**Figure supplement 3.** scRNAseq with TCR repertoire profiling of B6 and B6.*Sh2d1a*^-/- adult thymic γδ T ells.

γδNKT cells in our dataset, suggesting that they develop during a brief developmental window between E17 and D9 after birth.

A comparison of B6 adult thymic γδ T cells between B6 and B6.*Sh2d1a*^-/- mice recapitulated many of our observations of the neonatal thymus, although to a somewhat lesser degree. For example, we observed a small but significant decrease in the frequency of the CD24^high CD73^low adult thymic c1 and c2 clusters in B6.*Sh2d1a*^-/- mice that was coupled with an increased frequency of the CD24^pos CD73^+/- c0 cluster (*Figure 4—figure supplement 3*). Like the SAP-dependent E17 c2 and D9 c1 clusters, the adult thymus c1 and c2 clusters were enriched in γδT17-associated *Blk*, *Etv5*, *Sox13*, *Maf*, and *Rorc* (mostly in c2) indicating that SAP regulates this γδ T cell thymic developmental checkpoint at all ages. Likewise, both the adult thymus c0 and the neonatal c4 and c5 clusters that were enriched in *Ms4a4b*, *Ms4a6b*, *Dgka*, *Klf2*, and *S1pr1* exhibited increased frequencies in B6.*Sh2d1a*^-/- mice. Just as we observed in neonatal thymic γδ T cells, we noted a near complete loss of mature γδNKT cells (*Zbtb16*, *Icos*, and *Il4*-enriched γδT2 c8 cells and *Tbx21*, *Slamf7*, and *Il2rb*-enriched γδT1 c11 cells) as well as a decreased frequency of mature non-γδNKT γδT1 c10 cluster cells in B6.*Sh2d1a*^-/- adult thymus (*Figure 4—figure supplement 3*). Unlike the E17 and neonatal thymic datasets, however, the frequency of γδ T cells with an αβ T cell-like signature in the adult thymic γδ T cell dataset was minimal. Collectively, these data supported the presence of multiple SAP-dependent developmental changes during neonatal and adult γδ T cell development—a reduction in the CD24^pos CD73^neg SLAMF1^+SLAMF6^+ cluster enriched in γδT17 development-associated transcripts, an increased frequency of immature *S1pr1*-enriched γδ T cells, and decreased frequencies of both γδNKT cells (γδT1 and γδT2) and non-γδNKT γδT1 cells.

## SAP shapes the thymic γδ TCR repertoire

Given our observation of altered γδ TCR clonotype distribution among E17 γδ T cells and the substantial changes in clustering we observed in SAP-deficient mice, we were next interested in determining the effect of SAP deficiency on the neonatal and adult γδ TCR repertoires. A comparison of the frequencies of the top 100 neonate γδ TCR clonotypes between B6 and B6.*Sh2d1a*^-/- mice revealed significant changes in the overall frequency of several TRGV1 and TRGV4 clonotypes (*Figure 5A*). Consistent with a SAP-dependent loss of γδNKT, we observed a profound loss of TRGV1 clonotypes utilizing the canonical TRAV15N-1 and TRAV15-1/TRDV6-1 γδNKT cell TCRs (*Figure 5B*). Loss of these clonotypes was directly tied to the significant decrease of the γδT1 c8 and γδT2 c13 clusters, in which nearly all these clonotypes resided.

Within the B6 γδT2 c13 cluster the vast majority (~86%) of TCR sequences were highly restricted canonical γδNKT cell clonotypes (*Supplementary file 2*) and nearly all of these clonotypes were also present in the γδT1 c8 cluster, consistent with the ability of γδNKT cells to produce both IFN-γ and IL-4. In contrast, while the B6 γδT1 c8 cluster contained a large proportion of canonical γδNKT cell clonotypes (~46% of c8 TCRs), it also contained TCRs with a diverse pool of clonotypes that utilized TRGV1 (~70% of c8 TCRs), TRGV4, TRGV5, TRGV6, and TRGV7 that were present only in the c8 γδT1 cluster. Therefore, in neonates, the thymic γδT1 population is a mixture of γδNKT and 'non-γδNKT'

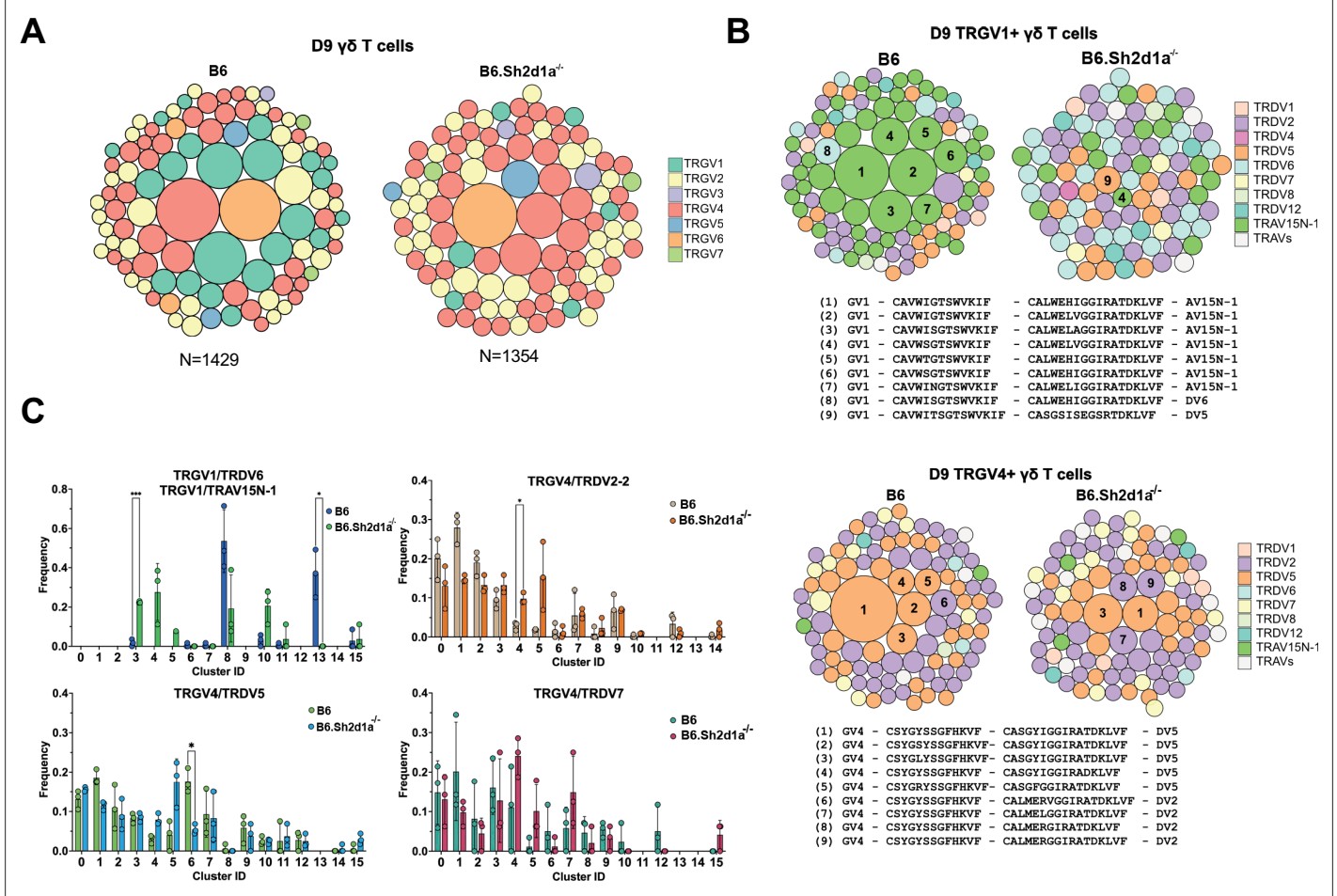

**Figure 5.** SAP shapes the neonatal γδ TCR repertoire. (**A**) TCR clonotype bubble plots depicting the top 100 B6 and B6.*Sh2d1a*-/- TCR clonotypes among D9 thymic γδ T cells. Each bubble represents a unique clonotype, is sized according to its frequency as a percentage of all γδ T cells *n* = 1429 B6, 1354 B6.*Sh2d1a*-/- cells, and is colored based on specific TRGV chains utilized. (**B**) TCR clonotype bubble plots depicting the top 100 B6 and B6.*Sh2d1a*-/- TRGV1 (*above*) and TRGV4 (*below*) clonotypes among D9 γδ T cells. Bubble size indicates clonotype frequency as a percentage of TRGV1+ (*n* = 198 B6, 98 B6.*Sh2d1a*-/-) or TRGV4+ (*n* = 646 B6, 669 B6.*Sh2d1a*-/-) cells, and colors correspond to specific TRDV chains. Selected clonotypes are numbered and their corresponding CDR3γ and CDR3δ sequences are displayed below. (**C**) Distribution of selected TRGV/TRDV clonotypes among B6 and B6.*Sh2d1a*-/- D9 thymic γδ T cell clusters. Data are representative of 3 mice/strain, *p ≤ 0.05, ***p ≤ 0.001, two-way ANOVA, Sidak multiple comparisons test.

The online version of this article includes the following source data for figure 5:

**Source data 1.** TCR clonotype distribution among neonatal B6 and B6.*Sh2d1a*-/- gamma delta T cells.

γδT1 clonotypes, and each of these clonotype pools is SAP-dependent. As noted above, these findings were largely reproduced in the adult thymus where the γδNKT clonotypes associated with the c8 and c11 clusters were virtually absent in B6.*Sh2d1a*-/- mice, and where the diverse pool of γδT1 c10 clonotypes was significantly diminished (*Supplementary file 2*).

Interestingly, a closer examination of the neonatal thymic TRGV4 clonotypes revealed a significant decrease in the frequencies of γδT17-associated clonotypes with a germline-encoded TRDV5 (*Kashani, 2015*; *Figure 5B*), some of which were identical to those that were redistributed to the αβ T cell-like c6 cluster in E17 γδ T cells (*Figure 2E*). When these clonotypes were mapped to their D9 clusters, we noted a significant decrease in the frequency of TRGV4/TRDV5 clonotypes in the mature γδT17 c6 cluster (*Figure 5C*). We note that while we observed a trend toward an increased frequency of TRGV4/TRDV5 and TRGV4/TRDV2 clonotypes in the *S1pr1*+ c4/c5 clusters, analysis of these sequences revealed that they largely exhibited diverse CDR3 rather than the restricted CDR3 observed in the γδT17 c6 cluster (*Supplementary file 2*). Taken together, these data indicated that

the immature E17 γδ TCR clonotypes that were redirected to the αβ T cell pathway in B6.*Sh2d1a*⁻/⁻ mice were underrepresented among the mature B6.*Sh2d1a*⁻/⁻ neonatal γδ T cells.

## Restricted TCR repertoire in innate-like γδT1 as well as γδT17 in the periphery

We previously demonstrated that SAP-deficiency results in a significant impairment in γδ T cell function in the periphery (*Dienz et al., 2020*). Specifically, we observed that γδ T cell IL-17 production is impaired in the lungs of young mice, but that this had largely corrected itself as the mouse reached maturity. In contrast, we observed that both Vγ1 and Vγ4 IFN-γ production were significantly impaired in B6.*Sh2d1a*⁻/⁻ adult lung and spleen. To better understand how SAP-dependent alterations in thymic development ultimately affected the peripheral γδ T cell compartment, we compared B6 and B6.*Sh2d1a*⁻/⁻ lungs using single-cell RNAseq with V(D)J profiling.

Our initial annotation of the B6 lung γδ T cell transcriptome revealed the presence of 11 clusters of the lung γδ T cells that we annotated based on cluster-specific gene expression and TRGV and TRDV usage (*Figure 6A*). Based on their expression of signature cytokines and transcription factors, we identified 2 γδT17 clusters (c1 and c4), 4 γδT1 clusters (c2, c3, c7, and c9), 1 γδTNKT-like cluster (c10) with very few cells, as well as a *Cd24*-enriched cluster (c5) (*Figure 6B, C, Supplementary file 5*). Consistent with previous observations, we noted that the c1 and c4 γδT17 clusters were enriched in *Slamf1* and that the c2/c3/c7/c9 γδT1 clusters were enriched in *Slamf7* and *Slamf6* (*Figure 6B*). Indeed, flow cytometric analysis revealed that SLAMF7 expression was restricted to the innate-like lung CD44⁺CD45RB⁺γδT1 cell population, and that γδT1 cells are SLAMF1⁻F6⁺F7⁺, while the CD44⁺CD45RB⁻ γδT17 subsets are SLAMF1⁺F6⁻F7⁻ (*Figure 6D*). Further annotation using V(D)J profiling revealed that the c1 cluster corresponded to Vγ4 γδT17 (primarily TRGV4/TRDV5 and TRGV4/TRDV2-2) while the c4 cluster corresponded to lung Vγ6 (TRGV6/TRDV4) (*Figure 6E, Supplementary file 6*). The c10 cluster was enriched in *Zbtb16*, *Il4*, and *Icos* (*Figure 6B, C*) as well as TRGV1/TRAV15N-1 and TRGV1/TRDV6 sequences (*Figure 6E, Supplementary file 6*) consistent with γδNKT cells. We did note, however, that the TRAV15N-1 and TRDV6 sequences in the c10 cluster were, for the most part, non-canonical γδNKT cell clonotypes, suggesting a degree of flexibility in the TRAV15N-1 and TRDV6 CDR3 sequences that can be utilized by γδNKT cells.

Interestingly, while the γδT1 c2/c3/c7/c9 clusters contained primarily Vγ1 and Vγ4 T cells as expected, we noted that Vγ4 γδT1 cells exhibited a striking preference in the usage of TRAV13-4(DV7), and that TRGV4/TRDV7 pairings accounted for 80% of all Vγ4 clonotypes in the γδT1 clusters (*Figure 6E, Supplementary file 6*). Analysis of the γδT1 TRAV13-4/DV7 CDR3 sequences revealed a high level of diversity that showed no evidence of clonal expansions (*Figure 6F*). This was in contrast to the highly restricted Vγ6 repertoire in the c4 γδT17 cluster, as well as the restricted Vγ4 repertoire in the c2 γδT17 cluster that was dominated by semi-invariant TRDV5 and TRDV2-2 sequences (*Figure 6F*). As no Vδ7-specific antibody was available, we independently confirmed this observation using single-cell PCR to assess the lung Vγ4 CDR3δ sequences from 13 individual mice. These data demonstrated that, other than TRDV2-2 and TRDV5, TRDV7 is the only other TCR delta chain used at an appreciable frequency by Vγ4 T cells in the B6 mouse lung and that it exhibits significant diversity in its CDR3 (*Figure 6—figure supplement 1*). Vγ1 γδT1 cells, in contrast, exhibited a more mild pairing preference where 44% of the γδT1 clonotypes utilized TRAV15N-1, TRAV15-1/DV6-1, or TRAV15D-2/DV6D-2 and 13% utilized TRAV13-4/DV7 (*Supplementary file 6*). Together these data formed a comprehensive map of B6 lung γδ T cell transcriptomes and revealed that lung innate-like γδT1, like γδT17 and γδNKT, are all characterized by preferential TCR γ- and δ-chain pairings. This was especially notable for Vγ4 and our data suggested that Vγ4 γδT1 exhibited a distinct preference for the utilization of TRAV13-4(DV7) chains with diverse CDR3.

## SAP regulates peripheral Vγ4/Vδ7 γδT1 cells and shapes the lung γδT17 repertoire

Next, we assessed the effect of SAP deficiency on the lung γδ TCR repertoire by comparing B6 and B6.*Sh2d1a*⁻/⁻ lung γδ T cells (*Figure 7A*). Consistent with our previous observation that lung γδ T cell IFN-γ production is deficient in *Sh2d1a*-deficient mice (*Dienz et al., 2020*), we observed a decrease in the frequency of the γδT1 c2, c3, and c7 clusters (*Figure 7B*). Moreover, we observed a striking decrease in the frequency of TRGV4/TRDV7-expressing cells in the lungs of B6.*Sh2d1a*⁻/⁻ mice

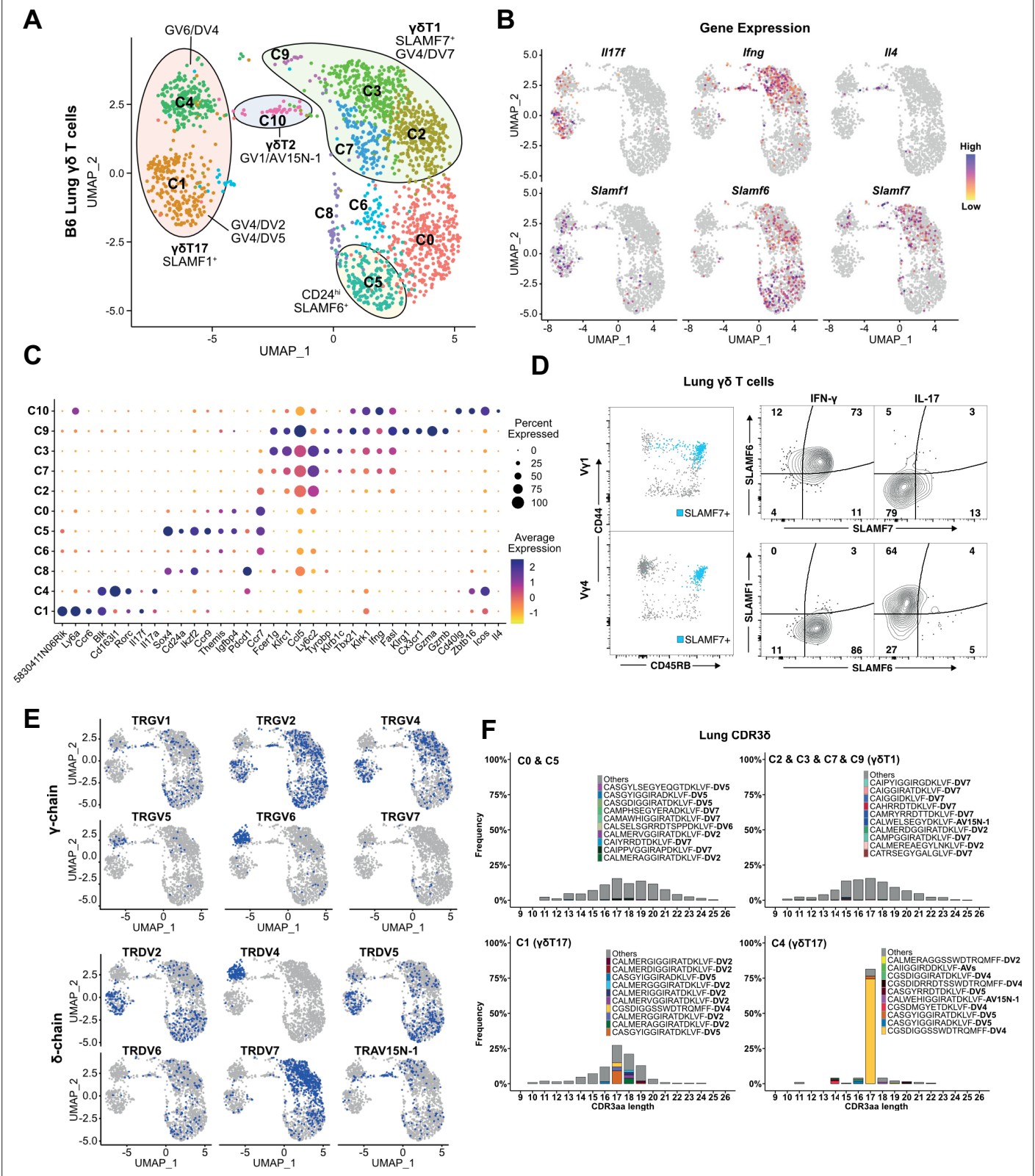

**Figure 6.** Restricted TCR repertoire in peripheral innate-like γδT1 and γδT17 subsets. (**A**) Uniform manifold approximation and projection (UMAP) representation of B6 adult lung γδ T cells. Clusters are annotated based on gene expression and TCR profiling data. (**B**) Feature plot displaying gene expression profiles of selected genes among individual B6 lung γδ T cells, color-coded based on gene expression levels. (**C**) Dot plot indicating scaled expression levels of selected genes in B6 lung γδ T cells, with dot size representing the fraction of cells within each cluster expressing the marker. (**D**)

*Figure 6 continued on next page*

*Figure 6 continued*

SLAMF6 and SLAMF7 co-expression marks CD44⁺CD45RB⁺ IFN-γ-producing cells in the periphery. Representative dot plots (left) of CD44 and CD45RB expression on lung Vγ1 (top) and Vγ4 (bottom) γδ T cells. SLAMF7-expressing cells are highlighted in blue. Representative contour plots (right) of SLAMF1, SLAMF6, and SLAMF7 expression on IFN-γ⁺ and IL-17⁺ lung γδ T cells. (**E**) UMAP representation of B6 lung γδ T cells indicating selected TRG and TRD chain V-segment usage (blue). (**F**) Amino acid length distribution of lung CDR3δ clonotypes among immature C0/C5, γδT1 C2/C3/C7/C9, γδT17 C1, and γδT17 C4 clusters. The top 10 clonotypes in each group are color-coded, while all other clonotypes are shown in gray.

The online version of this article includes the following figure supplement(s) for figure 6:

**Figure supplement 1.** Lung Vγ4 exhibit a high usage of TRDV7 chains with a diverse CDR3.

(***Figure 7C, D***, ***Supplementary file 6***), and we noted that we found no evidence of a compensatory increase in the frequency of other TRDV-expressing lung Vγ4 γδT1 cells in the B6.*Sh2d1a⁻/⁻* lung. We independently confirmed this SAP-dependent decrease in TRAV13-4/TRDV7 in both lung and spleen using qPCR (***Figure 7—figure supplement 1***), and we demonstrated a significant decrease in the number of CD44⁺SLAMF7⁺ Vγ4 T cells in both the lungs and spleen of B6.*Sh2d1a⁻/⁻* mice (***Figure 7E***, ***Figure 7—figure supplement 1***). The decreased lung γδT1 did not appear to be due to an SAP-dependent decrease in homeostatic proliferation, nor to increased apoptosis, since we observed no significant change in lung γδ T cell BrdU incorporation or Annexin V staining in B6.*Sh2d1a⁻/⁻* mice (***Figure 7—figure supplement 1***).

In addition to the decrease in lung γδT1, we also noted a loss of the c10 γδT2 cluster consistent with the enrichment in SAP-dependent γδNKT TCRs in this cluster, and we observed an increased frequency of the Vγ4 γδT17 c1 cluster (***Figure 7B***). Given our previous observation of a SAP-dependent decrease in lung γδ T cell IL-17 production in young, but not adult mice (***Dienz et al., 2020***), and our findings here of SAP-dependent alterations in γδT17 clonotypes, we were interested in determining whether the adult lung γδT17 repertoire was altered in SAP-deficient mice. Indeed, a comparison of the lung Vγ4 c1 γδT17 CDR3 sequences between B6 and B6.*Sh2d1a⁻/⁻* mice revealed significant alterations in the γδT17 c1 cluster TCR repertoire. Specifically, we noted a decreased frequency of TRGV4 chains that exhibited a germline-encoded CDR3γ (CSYGYSS) and an increased frequency of TRGV4 chains with a CDR3γ exhibiting evidence of N/P additions (CSYG(X)YSS). These changes were associated with a significant shift in the usage of TRDV2-2 CDR3δ sequence usage (***Figure 7F***) and an altered c1 TCR clonotype frequency in B6.*Sh2d1a⁻/⁻* mice (***Figure 7G***). Although the number of germline-encoded invariant TRGV4/TRDV5 clonotypes (clonotype 1 in ***Figure 6B***) was diminished in B6.*Sh2d1a⁻/⁻* lung γδ T cells (***Figure 7G***, ***Supplementary file 6***), the low numbers of cells available for analysis precluded us from making strong conclusions regarding the loss of any one particular clonotype. Taken together, these data suggest that Vγ4 γδT1 preferentially utilize the TRAV13-4(DV7) chain in the lung and that the SAP-dependent decrease in peripheral γδ T cell IFN-γ production is not due to dysregulated cytokine production, but rather due to a specific loss of innate-like γδT1 subsets. This finding, together with our finding that the adult lung γδT17 TCR repertoire is altered in SAP-deficient mice, suggests that SLAM/SAP signaling plays a role in shaping the γδ TCR repertoire.

## Discussion

A common denominator in most innate-like T cell developmental programming is a dependence on the SLAM/SAP signaling pathway (***Kreslavsky et al., 2009***; ***Dienz et al., 2020***; ***Legoux et al., 2019***; ***Griewank et al., 2007***). Within the γδ T cell lineage, SAP plays a critical role in the development of IFN-γ and IL-4-producing γδNKT cells (***Azuara et al., 1997***), as well as in γδT1 and γδT17 (***Dienz et al., 2020***), but exactly when during γδ T cell development SAP functions, and how it exerts these effects is unknown. Here, we provide evidence that SAP mediates its effects on immature, uncommitted thymic γδ T cells and that it works to simultaneously promote progression through the γδT17 developmental pathway while inhibiting the diversion of γδ T cells into the αβ T cell pathway. The redirection of γδ T cell clonotypes into the αβ T cell pathway at E17 is associated with a decreased frequency of these mature clonotypes in neonatal thymus, and an altered γδT17 TCR repertoire in the periphery. Finally, we identify Vγ4/Vδ7 T cells as a novel, SAP-dependent IFN-γ-producing Vγ4 γδT1 subset.

The developmental pathway(s) that gives rise to γδ T cell functional subsets remains unclear. γδ T cells develop at the DN2/DN3 stage after receiving a sufficiently strong γδ TCR-mediated signal (***Kreslavsky et al., 2008***). Numerous reports support a model where, within the continuum of signals

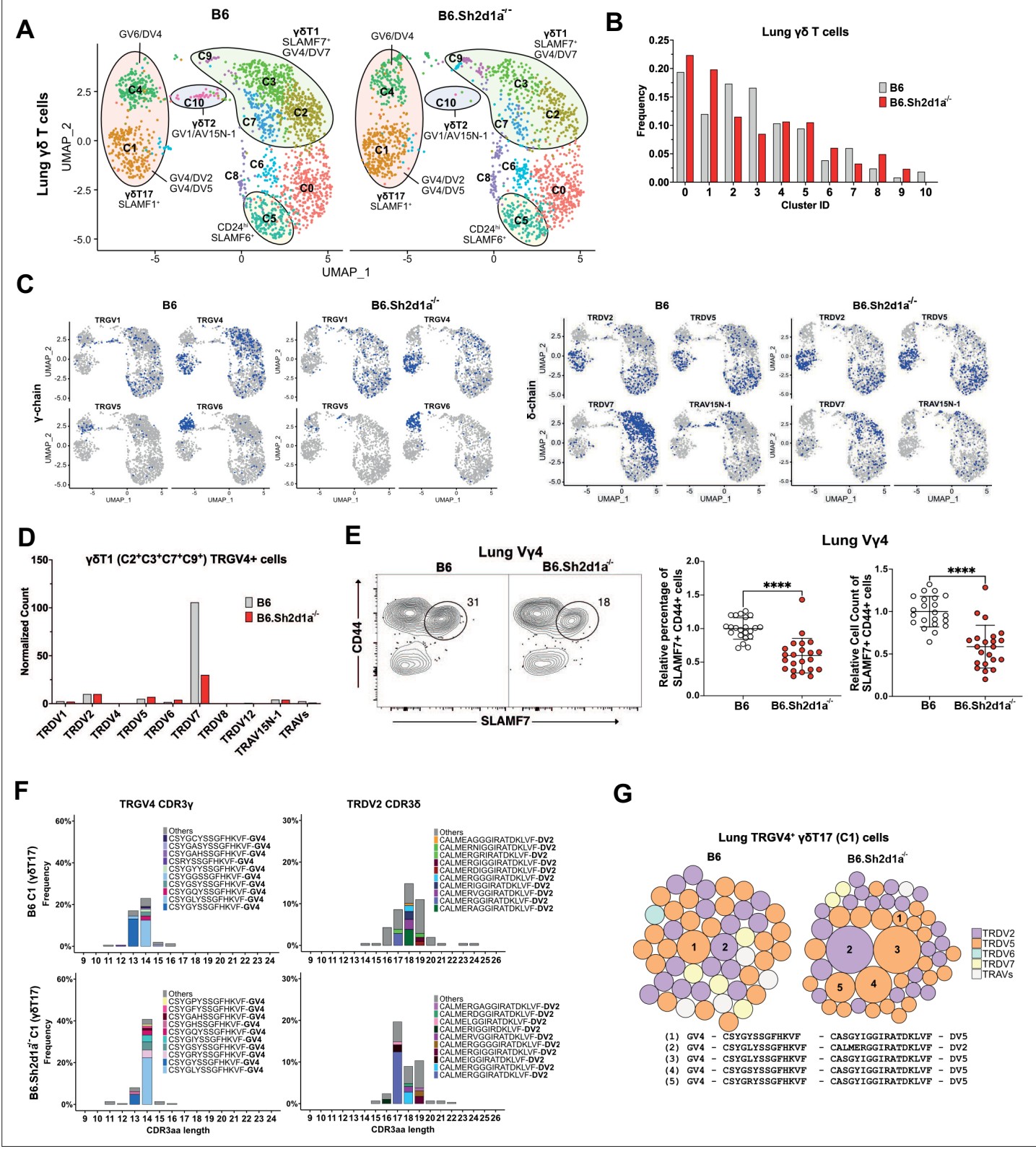

**Figure 7.** Specific decrease in Vγ4/Vδ7 γδT1 cells in the lungs of SAP-deficient mice. (**A**) Uniform manifold approximation and projection (UMAP) representation of B6 (left) and B6.*Sh2d1a*^-/- (right) lung γδ T cells, pooled from 3 mice per strain. (**B**) SAP-dependent variation in lung γδ T cell cluster frequencies. The frequency of each B6 and B6.*Sh2d1a*^-/- cluster as a percentage of all lung γδ T cells is shown. (**C**) UMAP representation of selected TRG and TRD chain V-segment usage (blue) among B6 and B6.*Sh2d1a*^-/- lung γδ T cells. (**D**) SAP-dependent decrease in lung γδT1 TRGV4/TRDV7.

*Figure 7 continued on next page*

*Figure 7 continued*

Normalized TRDV counts in B6 and B6.*Sh2d1a*[-/-] lung TRGV4[+] γδT1 clusters (C2/C3/C7/C9). (**E**) SAP-dependent decrease in SLAMF7[+] lung Vγ4 T cells. Representative contour plots of CD44 and SLAMF7 expression in B6 (left) and B6.*Sh2d1a*[-/-] (right) lung Vγ4 γδ T cells are shown at *left*. Relative frequencies and counts of CD44[+]SLAMF7[+] lung Vγ4 cells is shown at *right*. Data represent the cumulative data from five independent experiments, ****p < 0.0001 using unpaired *t*-test. (**F**) SAP-dependent skewing of the lung γδT17 TCR repertoire. Amino acid length distributions of B6 and B6.*Sh2d1a*[-/-] TRGV4[+] CDR3γ (left) and TRDV2[+] CDR3δ (right) sequences in the lung Vγ4 γδT17 C1 cluster. The top 10 clonotypes are color-coded and all other clonotypes are shown in gray. Bars represent the frequency of γδ T cells in the c1 cluster. (**G**) TCR clonotype bubble plots depicting the top 50 lung TRGV4 clonotypes in the C1 γδT17 cluster. Bubble size indicates clonotype frequency as a percentage of C1 Vγ4 (n = 62 B6, 104 B6.*Sh2d1a*[-/-]) T cells, and colors correspond to specific TRDV chains. Selected clonotypes are numbered and their corresponding CDR3γ and CDR3δ sequences are displayed below.

The online version of this article includes the following source data and figure supplement(s) for figure 7:

**Figure supplement 1.** Significant and specific reduction of peripheral TRGV4/TRDV7[+] γδT1 in B6.*Sh2d1a*[-/-] mice**.**

**Figure supplement 1—source data 1.** Characterization of Vγ4 γδT1 in B6 and B6.*Sh2d1a*[-/-] periphery.

that can drive γδ T cell development, it is the relatively weaker TCR signals that promote γδT17 programming while the stronger TCR signals foster γδNKT and γδT1 programming (***Sumaria et al., 2017***; ***Jensen et al., 2008***; ***Kreslavsky et al., 2009***; ***Sumaria et al., 2021***; ***Fahl et al., 2018***; ***Verykokakis et al., 2010***) Despite considerable effort, the precise thymic developmental stages at which γδ T cell lineage diversification occurs remains unclear. Recent studies examining this issue have demonstrated that effector programming can occur at a CD44[neg]CD45RB[neg] stage (***Sumaria et al., 2017***; ***Sumaria et al., 2021***), and that c-Maf-directed γδT17 programming occurs at a CD45RB[low] stage (***Zuberbuehler et al., 2019***). In addition, by altering TCR signal strength in a transgenic model, Chen et al., defined a *Ccr9*-enriched cluster as a point of effector lineage diversification (***Chen et al., 2021***). Our findings here suggest that SAP influences γδT17 developmental programming during the immature uncommitted CD24[hi]CD73[neg] stage, either during or soon after *Il2ra* (CD25) downregulation. The SAP-dependent c2 cluster we define here was highly enriched in numerous genes (e.g., *Sox13* (***Malhotra et al., 2013***), *Blk* (***Laird et al., 2010***), *Maf* (***Zuberbuehler et al., 2019***), *Etv5* (***Jojic et al., 2013***), *Sh2d1a Dienz et al., 2020***) previously associated with γδT17 development as well as *Ccr9*, and the cell surface phenotype of c2 γδ T cells is SLAMF1[pos]SLAMF6[pos]CD44[low/high]CD45RB[low]. Altogether, these data suggest that the SAP-dependent c2 cluster is identical to these previously defined developmental checkpoints. Given that these previous studies have demonstrated that this developmental checkpoint is influenced by TCR signaling, it is interesting to note that SLAM/SAP signaling has previously been demonstrated to foster semi-invariant NKT cell differentiation through its ability to regulate TCR signal strength (***Lu et al., 2019***).

To that end, we noted that the SAP-dependent c2 cluster in E17 γδ T cells was enriched in *Blk, Lck,* and *Fyn*, Src tyrosine kinase family members that have all been implicated in γδT cell development (***Laird et al., 2010***; ***Laird and Hayes, 2010***). Decreased expression of *Blk*, which is known to promote γδT17 development (***Laird et al., 2010***), in B6.*Sh2d1a*[-/-] γδ T cells suggests one possible mechanism through which SAP may regulate γδT17 development via TCR signaling. The mechanism, however, through which SAP regulates *Blk* gene expression remains unclear. Since SAP is known to regulate conventional αβ T cell signaling through its ability to recruit FYN (***Davidson et al., 2004***; ***Chan et al., 2003***) and LCK (***Simarro et al., 2004***; ***Katz et al., 2014***) to SLAM family members, it seems reasonable to hypothesize that SAP could regulate TCR signaling by recruiting one or more of these Src kinase family members to the γδ TCR signalosome.

Our finding that SAP inhibited the number of γδ T cells that enter the αβ T cell developmental pathway is reminiscent of previous work using transgenic TCR models demonstrating that a reduction of TCR signal strength results in a redirection of γδ T cells to the αβ T cell lineage (***Haks et al., 2005***; ***Hayes et al., 2005***). Fahl et al. recently demonstrated that both entry into the γδ T cell lineage as well as γδ T cell effector programming involved repression of TCF1 expression by TCR-driven increases in *Id3* (***Fahl et al., 2021***). Consistent with these results, we found that *Tcf7* (encoding TCF1) transcript levels were highest in the αβ T cell-like c6 cluster, and we note that *Tcf7* transcripts levels were noticeably higher in the *Il2ra*-enriched post-selection c5 cluster in B6.*Sh2d1a*[-/-] γδ T cells. *Id3* expression, however, did not appear to be appreciably altered in SAP-deficient mice. Whether SAP-mediated signals could alter γδ T cell *Tcf7* expression through a TCR-independent mechanism remains an open question.

In addition to the considerable evidence supporting a role for TCR signal strength in γδ T cell developmental programming, there is increasing support for a model in which γδT17 programming occurs earlier in development, prior to expression of the γδ TCR. Indeed, evidence that some DN1 thymocytes possess a SOX13[hi] IL-17-like phenotype and that Vγ4 γδT17 arise from DN1 precursors (*Spidale et al., 2018*; *Oh et al., 2023*) indicates the existence of such a TCR-independent developmental programming stage. In consideration of such a model, it may be possible that SLAM/SAP-mediated signals could differentially regulate the survival and/or fitness of the functionally pre-programmed γδ T cell subsets. Such a function would be consistent with the role of SLAMF6 and SAP in promoting reactivation-induced cell death in conventional αβ T cells (*Katz et al., 2014*; *Snow et al., 2009*).

While our data indicate that SAP regulates the diversion of at least some TCR clonotypes into the αβ TCR pathway, the fate of those cells is unclear. Certainly, the fact that the clonotypes that were diverted to the αβ T cell pathway in E17 thymus are under-represented in the mature D9 thymus γδT17 clusters suggests that this diversion has significant consequences on the γδ T cell receptor repertoire, but whether these diverted γδ T cells contribute to the αβ T cell compartment is unclear. Recently, a population of hybrid αβ-γδ T cells was described in both mouse and human (*Edwards et al., 2020*) that responds to peptide/MHC stimulation like conventional αβ T cells, as well as to IL-1 and IL-23 which induce both IFN-γ and IL-17 production (*Edwards et al., 2020*). In the mouse, at least some of these cells are Vγ4/Vδ5, Vγ4/Vδ4, or Vγ4/Vδ7 and are present in the embryonic thymus, raising the possibility that γδ T cells diverted to the αβ T cell pathway in B6.*Sh2d1a*^-/- mice contribute to this αβ-γδ T cell population. While we have not observed a significant increase in the overall numbers of αβ-γδ T cells in B6.*Sh2d1a*^-/- thymus (not shown), a direct comparison of B6 and B6.*Sh2d1a*^−/−αβ-γδ TCRs may be necessary to see such an effect.

Our previous observations that SAP deletion did not completely abolish the development of IL-17- and IFN-γ-producing γδ T cells (*Dienz et al., 2020*) suggested the possibility that SAP may be critical for the development of additional γδ T cell clonotypes other than the Vγ1Vδ6.3 subset. Therefore, we compared the SAP-dependent and -independent γδTCR repertoires using paired TCR clonotype analysis, which allowed us to track clonotype fate during development. Our findings not only confirmed previous results demonstrating highly restricted γδT17 (*Wei et al., 2015*; *Kashani, 2015*), and γδNKT (*Azuara et al., 1997*) TCR repertoires, they also revealed that the vast majority of Vγ4 γδT1 subsets in the periphery preferentially utilize TRGV4/TRAV13-4(DV7) clonotypes, thereby further highlighting the strong links between γδ T cell function and the utilization of specific TCRs (*O'Brien and Born, 2010*). The reason for this preferential TRDV usage in Vγ4/Vδ7 γδT1 is unclear. While the TRAV13-4(DV7) CDR3 sequences were diverse and did not suggest evidence of clonal expansion, it may be possible that TRDV-encoded sequence motifs could be required to interact with a ligand, similar to the way that TRGV-encoded sequences mediate recognition of butyrophilins (*Rigau et al., 2020*; *Melandri et al., 2018*).

Exactly when during development SAP exerts its effects on Vγ4/Vδ7 T cells is still somewhat unclear. We found comparatively low numbers of TRGV4/TRDV7 γδ T cells in embryonic, neonatal, or adult thymus, and those that were identified were generally found in the immature clusters. The increased frequency in B6.*Sh2d1a*^-/- E17 thymus of TRGV4/TRDV7 γδ T cells in the *Il2ra*^+ c5 cluster coupled with their decreased frequency in the emigrating *S1pr1*^+ c4 cluster suggests a model where SAP promotes progression through the early stages of development through thymic egress. SAP deficiency, therefore, could result in lower Vγ4/Vδ7 output. However, this model is at odds with our finding that the frequency of *S1pr1*^+ clusters in neonatal thymus is increased in B6.*Sh2d1a*^-/-, and that these clusters contained increased numbers of TRGV4/TRDV7 γδ T cells. A possible explanation could be found in a previous report demonstrating that a Vγ4/Vδ7 subset increases in frequency in the periphery of B6 mice over time and undergoes extrathymic expansion (*Sperling et al., 1997*). Our finding here that the Vγ4/Vδ7 T cells represent an SAP-dependent innate-like γδT1 subset raise the possibility that SAP regulates the homeostatic expansion of this subset after it leaves the thymus. The mechanism, however, through which SAP is regulating the number of Vγ4/Vδ7 T cells remains unclear, as we have so far observed no evidence of changes in homeostatic proliferation or apoptosis of these cells in the lungs.

We previously demonstrated that the SAP-dependent impairment in γδT17 function in the neonatal periphery largely corrects itself as mice mature (*Dienz et al., 2020*). This suggested the possibility that SAP-dependent γδT17 clonotypes originating in the embryonic thymus were being replaced over

time by SAP-independent γδT17 clonotypes. Although our data certainly point to a SAP-dependent decrease in a germline-encoded Vγ4/Vδ5 clonotype in neonatal thymus, the low numbers of this clonotype in the B6 lung makes it difficult to draw definitive conclusions in the periphery. Still, we did observe a role for SAP in shaping the peripheral γδT17 repertoire, as we noted a decreased frequency of germline-encoded TRGV4 sequences (so-called 'GYSS'; *Sim and Augustin, 1991*) coupled with an increased frequency of TRGV4 sequences with more diverse CDR3 (GXYSS) in the periphery. Although the potential significance of this change on γδ T cell function is unclear, it does support a model in which SAP regulates γδT17 developmental programming during embryonic thymic development, where the low expression of terminal deoxynucleotide transferase results in a high frequency of germline-encoded V–J junctions.

In summary, our findings reveal the presence of multiple SAP-dependent developmental checkpoints during the very early stages of γδ T cell development in the thymus, indicating a more complex role for this signaling pathway in γδ T cell developmental programming than previously appreciated. Indeed, our data indicate that SAP functions to regulate entry into the γδ and αβ T cell developmental pathways, and we make the striking observation that nearly all Vγ4 γδT1 cells in the periphery preferentially use Vδ7 chains and that it is a decrease in the Vγ4/Vδ7 subset that, in part, underlies the SAP-dependent decrease in γδ T cell IFN-γ production. Altogether, these findings suggest that SAP regulates the balance of innate-like γδT1/γδT2/γδT17 subsets during thymic development and in the periphery.

# Materials and methods

**Key resources table**

| Reagent type (species) or resource | Designation | Source or reference | Identifiers | Additional information |
|---|---|---|---|---|
| Strain, strain background (*Mus musculus*) | C57BL/6J | Jackson Laboratory | Cat#: 000664 | |
| Strain, strain background (*Mus musculus*) | B6.129S6-Sh2d1a$^{tm1Pls/J}$ | Jackson Laboratory | Cat#: 025754 | |
| Biological sample (*Mus musculus*) | Thymus, spleen, lung | This paper | | Freshly isolated tissue |
| Antibody | Anti-mouse CD16/32, Fc block (rat monoclonal) | Biolegend | Cat#: 101301 | (1:200) |
| Antibody | Anti-CD4-BUV395 (rat monoclonal) | Thermo Fisher | Cat#: 363-0042-82 | (1:500) |
| Antibody | Normal Rat Serum (IgG) | Stem Cell Technologies | Cat#: 13551 | (1:200) |
| Antibody | Anti-CD11b-Alexa Fluor 647 (rat monoclonal) | Biolegend | Cat#: 101220 | (1:1000) |
| Antibody | Anti-CD11c-Alexa Fluor 647 (rat monoclonal) | Biolegend | Cat#: 117314 | (1:200) |
| Antibody | Annexin V Conjugate | Thermo Fisher | Cat#: 35111 | (1:20) |
| Antibody | Anti-CD19-Alexa Fluor 647 (rat monoclonal) | Biolegend | Cat#: 115525 | (1:500) |
| Antibody | Anti-CD24-SB600 (rat monoclonal) | Thermo Fisher | Cat#: 63-0242-82 | (1:200) |
| Antibody | Anti-CD24-BV605 (rat monoclonal) | Biolegend | Cat#: 101827 | (1:400) |
| Antibody | Anti-CD25-PE-Dazzle594 (rat monoclonal) | Biolegend | Cat#: 102048 | (1:500) |
| Antibody | Anti-CD25-Alexa Fluor 488 (rat monoclonal) | Biolegend | Cat#: 102017 | (1:800) |
| Antibody | Anti-CD27-PE-Dazzle594 (hamster monoclonal) | Biolegend | Cat#: 124228 | (1:400) |
| Antibody | Anti-CD44-BV510 (rat monoclonal) | Biolegend | Cat#: 103044 | (1:400) |
| Antibody | Anti-CD45RB-APC-Cy7 (rat monoclonal) | Biolegend | Cat#: 103310 | (1:800) |
| Antibody | Anti-CD73-APC-F750 (rat monoclonal) | Biolegend | Cat#: 127222 | (1:400) |
| Antibody | Anti-CD150-BV650 (rat monoclonal) | Biolegend | Cat#: 115931 | (1:400) |
| Antibody | Anti-CD196-BV785 (hamster monoclonal) | Biolegend | Cat#: 129823 | (1:200) |

*Continued on next page*

*Continued*

| Reagent type (species) or resource | Designation | Source or reference | Identifiers | Additional information |
|---|---|---|---|---|
| Antibody | Anti-CD319-APC (rat monoclonal) | Biolegend | Cat#: 152004 | (1:800) |
| Antibody | Anti-cMAF-Alexa488 (mouse monoclonal) | Thermo Fisher | Cat#: 53-9855-82 | (1:100) |
| Antibody | Anti-IFNγ-PE (rat monoclonal) | Biolegend | Cat#: 505808 | (1:200) |
| Antibody | Anti-IL17a-BV605 (rat monoclonal) | Biolegend | Cat#: 506927 | (1:200) |
| Antibody | Anti-IL4-PE-Cy7 (rat monoclonal) | Biolegend | Cat#: 504118 | (1:100) |
| Antibody | Anti-PLZF-PE (hamster monoclonal) | Biolegend | Cat#: 145804 | (1:100) |
| Antibody | Anti-T-bet-PE-Dazzle-594 (mouse monoclonal) | Biolegend | Cat#: 644828 | (1:200) |
| Antibody | Anti-TCRb-Alexa647 (hamster monoclonal) | Biolegend | Cat#: 109218 | (1:800) |
| Antibody | Anti-Vγ5-FITC (hamster monoclonal) | BD Biosciences | Cat#: 553229 | (1:300) |
| Antibody | Anti-Vδ6.3-R718 (hamster monoclonal) | BD Biosciences | Cat#: 752197 | (1:250) |
| Antibody | Anti-RORγt-PE-CF594 (mouse monoclonal) | BD Biosciences | Cat#: 562684 | (1:200) |
| Antibody | Anti-TCRδ-BB700 (hamster monoclonal) | BD Biosciences | Cat#: 745818 | (1:100) |
| Antibody | Anti-Vγ1-BV421 (hamster monoclonal) | Biolegend | Cat#: 141116 | (1:300) |
| Antibody | Anti-CD8α-Alexa700 (rat monoclonal) | Thermo Fisher | Cat#: 56-0081-82 | (1:400) |
| Antibody | Anti-CD45-eFluor506 (rat monoclonal) | Thermo Fisher | Cat#: 69-0451-82 | (1:400) |
| Antibody | Anti-CD200-PerCP-eF710 (rat monoclonal) | Thermo Fisher | Cat#: 46-5200-82 | (1:500) |
| Antibody | Anti-CD278-SB645 (rat monoclonal) | Thermo Fisher | Cat#: 64-9942-82 | (1:1000) |
| Antibody | Anti-Ki67-PE-eFluor610 (rat monoclonal) | Thermo Fisher | Cat#: 61-5698-82 | (1:1000) |
| Antibody | Anti-Ly108-BUV615 (13G3 monoclonal) | BD Bioscience | Cat#: 751341 | (1:400) |
| Antibody | Anti-TCR-Vγ4-PE-Cy7 (hamster monoclonal) | Thermo Fisher | Cat#: 25-5828-82 | (1:400) |
| Antibody | Anti-S1P1 (rat monoclonal) | R&D Systems | Cat#: FAB7089P | (1:100) |
| Antibody | Anti-Blk (rabbit polyclonal) | Cell Signaling Technologies | Cat#: 3262S | (1:200) |
| Antibody | Anti-Vdelta1 (17D1) (rat monoclonal) | Willi Born; National Jewish Health, CO, USA | Hybridoma | |
| Sequence-based reagent | GDC1F | This paper | | Sequence is in Materials and methods |
| Sequence-based reagent | GC1R1 | This paper | | Sequence is in Materials and methods |
| Sequence-based reagent | GC1R2 | This paper | | Sequence is in Materials and methods |
| Sequence-based reagent | DC1R1 | This paper | | Sequence is in Materials and methods |
| Sequence-based reagent | GDC2F | This paper | | Sequence is in Materials and methods |
| Sequence-based reagent | GC2R1 | This paper | | Sequence is in Materials and methods |

*Continued on next page*

*Continued*

| Reagent type (species) or resource | Designation | Source or reference | Identifiers | Additional information |
|---|---|---|---|---|
| Sequence-based reagent | GC2R2 | This paper | | Sequence is in Materials and methods |
| Sequence-based reagent | DC2R1 | This paper | | Sequence is in Materials and methods |
| Commercial assay or kit | Fixable UV LIVE/DEAD Kit | Thermo Fisher Scientific | Cat#: L23105 | (1:400) |
| Commercial assay or kit | BD Horizon Brilliant Staining Buffer | BD Biosciences | Cat#: 566349 | |
| Commercial assay or kit | Invitrogen eBioscience Foxp3/Transcription Factor Fixation/Permeabilization Concentrate and Diluent | Thermo Fisher Scientific | Cat#: 50-112-9060 | |
| Commercial assay or kit | BrdU Flow Kit | BD Biosciences | Cat#: 559619 | |
| Commercial assay or kit | Fixable Viability Dye eFluor 780 | Thermo Fisher Scientific | Cat#: 65-0865-14 | (1:1000) |
| Commercial assay or kit | Binding Buffer for Annexin V (10×) | Thermo Fisher Scientific | Cat#: 00-0055-43 | |
| Chemical compound, drug | 5-bromo-deoxyuridine (BrdU) | Sigma-Aldrich | Cat#: 19-160 | |
| Software, algorithm | FlowJo | BD Biosciences | | |
| Software, algorithm | R | R Foundation | RRID:SCR_001905 | |

## Mice

C57BL/6J (B6) and B6.129S6.*Sh2d1a*-/- mice were purchased from the Jackson Laboratory (Bar Harbor, ME) and were housed and bred in a specific pathogen-free facility at the animal facility of the University of Vermont. For experiments using E17 mice, timed-pregnant matings were used. The presence of a vaginal plug indicated day 0 of pregnancy. The flow cytometry experiments included 8- to 13-week-old B6 and B6.*Sh2d1a*-/- mice (age- and sex-matched). All experimental procedures on animals were carried out with the approval of the University of Vermont Institutional Animal Care and Use Committee under Protocols PROTO202000073 and PROTO202000179.

## Tissue processing

Single-cell suspensions from the spleen and thymus were obtained by gently passing tissue through nylon mesh. Red blood cells (RBCs) were lysed with Gey's solution. Single-cell suspensions from the lungs were prepared as described (*Benoit et al., 2015*). Briefly, dissected lungs were finely minced in a digestion buffer containing DMEM, 1 mg/ml collagenase type IV (Gibco), and 0.2 mg/ml DNase (Sigma-Aldrich). After shaking at 200 rpm at 37°C for 20 min, the lung digests were triturated with a 16-gauge needle, followed by an additional 20 min shaking at 200 rpm at 37°C. Following the final trituration, RBCs were lysed with Gey's solution. Finally, the digested lung tissues were filtered through a 70-µm filter and washed in phosphate-buffered saline (PBS)/2% fetal bovine serum. Cells were counted on either a MACSQuant VYB or using a hemacytometer following trypan blue staining.

## Flow cytometry

Cells were incubated with LIVE/DEAD Fixable Blue Dead Cell stain (Thermo Fisher Scientific, Grand Island, NY) for 30 min at 4°C in PBS, after which they were resuspended in Fc-Block (BioLegend) containing PBS/2% (fetal calf serum) FCS/0.1% sodium azide buffer. After a 5- to 10-min incubation on ice, antibodies specific for cell surface markers were added, incubated for 30 min on ice, and washed in PBS/2% FCS/0.1% sodium azide buffer. Brilliant Violet Staining buffer (BD Biosciences) was used in experiments containing more than one Brilliant Violet, Brilliant UV, or SuperBright-conjugated Ab. Cells were fixed in PBS/1% paraformaldehyde buffer and stored at 4°C prior to data collection.

For nuclear staining, cells were permeabilized with Foxp3 transcription factor fixation/permeabilization buffer (Thermo Fisher) following surface staining. Cells were incubated overnight at 4°C, washed, and incubated with 25 µg/ml total rat IgG before adding Abs against transcription factors. After staining, cells were washed and resuspended in PBS/2% FCS/0.1% sodium azide buffer. For intracellular cytokine staining, cells were permeabilized with PBS/1% FCS/0.1% sodium azide/0.1%

saponin following surface staining and fixation. To reduce unspecific binding, 25 µg/ml total rat IgG was added prior to cytokine staining with fluorescent-labeled antibodies.

Flow cytometry data were collected on a Cytek Aurora (Cytek Biosciences) spectral cytometer, and data analysis was performed using FlowJo Software (BD Biosciences). In some cases, visualization of flow cytometry data was conducted by concatenation of individual .fcs files, followed by dimensionality reduction and clustering using UMAP and FlowSOM (*Van Gassen et al., 2015*) in FlowJo. Antibodies used in these experiments are anti-CD4 (RM4-5), anti-CD11b (M1/70), anti-CD11c (N418), anti-CD19 (1D3), anti-CD24 (M1/69), anti-CD25 (PC61), anti-CD27 (LG.3A10), anti-CD44 (IM7), anti-CD45RB (C363-16A), anti-CD73 (TY/11.8), anti-CD150 (TC15-12F12.2), anti-CD196 (29-2L17), anti-CD319 (4G2), anti-cMAF (sym0F1), anti-IFNγ (XMG1.2), anti-IL17a (TC11-18H10.1), anti-IL4 (11B11), anti-PLZF (9E12), anti-T-bet (4B10), anti-TCRβ (H57-597), anti-Vγ5 (536), and anti-Vδ6.3 (C504.17c) from BioLegend; and anti-RORγt (Q31-378), anti-TCRδ (GL3), anti-Vγ1 (2.11) from BD biosciences; and anti-CD8a (53–6.7), anti-CD45 (30-F11), anti-CD200 (OX90), anti-CD278 (7E17G9), anti-Ki-67 (SolA15), anti-Ly108 (13G3-19D), anti-Vγ4 (UC3-10A6) from eBiosciences; anti-S1P1 (FAB7089P) from R&D systems; anti-Blk (polyclonal) from Cell Signaling Technology; and anti-17D1 (a gift from Willi Born, with permission for Robert Tigelaar).

## In vivo proliferation and apoptosis

For in vivo proliferation analysis, each mouse was intraperitoneally injected with 1 mg of 5-bromo-deoxyuridine (BrdU) (Sigma-Aldrich) each day for 3 days. Lung cell suspensions were prepared as above, stained with surface markers, after which they were stained with anti-BrdU antibody according to the manufacturer's instructions (BrdU Flow Kit; BD Biosciences). For apoptotic cell detection, annexin V staining was performed. Briefly, surface staining was followed by dead cell staining with Fixable Viability Dye eFluor 780 dye. The cells were washed and resuspended in an annexin-binding buffer (10 mM 4-(2-hydroxyethyl)-1-piperazineethanesulfonic acid [HEPES], 140 mM NaCl, and 2.5 mM $CaCl_2$, pH 7.4). Finally, 1 µl of Annexin V Conjugate (Thermo Fisher Scientific, Grand Island, NY) was added to each tube, and cells were incubated for 15 min at room temperature prior to data collection.

## Bulk RNA sequencing and data analysis

For the bulk RNA sequencing, thymocytes from 10-day-old B6 (*n* = 4 male, *n* = 2 female) and B6.*Sh2d1a*$^{-/-}$ (*n* = 5 male, *n* = 1 female) mice were prepared by depleting CD4+ cells from thymocytes using the EasySepTM Mouse CD4 Positive Selection Kit II (STEMCELL Technologies). Cells were then incubated with Fixable Viability Dye eFluor 780 (Thermo Fisher Scientific, Grand Island, NY) for 30 min at 4°C in PBS, after which they were resuspended in Fc-Block (BioLegend) containing PBS/2% FCS. After washing, cells were stained with antibodies against CD11b (M1/70), CD19 (1D3), CD24 (M1/67), CD45 (30-F11), TCRβ (H57-597), and TCRδ (GL3) for 30 min at 4°C. Following surface staining, CD24$^{hi}$TCRδ+ cells were sorted from a CD11b$^-$CD19$^-$TCRb- population of live CD45$^+$ lymphocytes. Immature (CD24$^{high}$) thymocytes from both B6 and B6.*Sh2d1a*$^{-/-}$ mice were sorted directly into 350 µl of RLT lysis buffer (QIAGEN). RNA was extracted using the RNeasy Micro kit (QIAGEN) according to the manufacturer's instructions. After RNA quantification and quality assessment, cDNA libraries were prepared using SMARTer Stranded Total RNA-Seq Kit v3 - Pico Input Mammalian (Takara, Japan) according to the manufacturer's protocol. Following library clean-up and quantification, the libraries were sequenced (single-end 75 bp) on an Illumina HiSeq1500.

The Fastq files were checked for data quality using FastQC and MultiQC (*Ewels et al., 2016*) bioinformatics tools. Cutadapt (*Martin, 2011*) was used to filter out low-quality sequences and to remove adapter sequences from all reads. The expression of transcripts was quantified using the mapping-based mode of Salmon (*Patro et al., 2017*), which quasi-mapped RNAseq reads to an index created from a mouse reference transcriptome (GRCm39). The transcript abundance estimates were imported using tximport (*Soneson et al., 2015*) and differential gene expression (DE) analysis was performed using DEseq2 (*Love et al., 2014*) employing the Wald test of the negative binomial model coefficients to determine statistical significance. The results were analyzed based on Log$_2$ fold change (Log$_2$FC ≥0.5 or ≤–0.5) and adjusted p-value (p$_{adj}$ ≤ 0.0001) to determine whether a gene was differentially expressed.

## Cell sorting and library preparation for CITE-seq and V(D)J enrichment

For thymus single-cell transcriptome analysis, we used embryonic day 17 (E17) B6 (*n* = 4) and B6.*Sh2d1a*^-/- (*Sh2d1a*^-/-) (*n* = 4) mice, 9-day-old (D9) B6 (*n* = 3) and B6.*Sh2d1a*^-/- (*n* = 3) mice, and 6-week-old B6 (*n* = 3 male) and B6.*Sh2d1a*^-/- (*n* = 2 male, *n* = 1 female) mice. For single-cell analysis of adult lung γδ T cells, we used 13-week-old B6 and B6.*Sh2d1a*^-/- littermates (pooled cells from two males and one female mouse in each group).

Analysis of gene expression, surface protein expression, and TCR V(D)J repertoire was performed using the 10X Genomics Chromium Next GEM Single Cell 5′ V(D)J with feature barcoding kit (version 1.1). To sort cells for scRNAseq, thymocytes were stained with anti-TCRδ-PE (GL3) antibody and enriched using anti-PE MicroBeads kit (Miltenyi Biotec). The exception was E17 thymus which did not undergo TCRδ enrichment. Following live/dead staining with Fixable Viability Dye eFluor 780 (Thermo Fisher Scientific, Grand Island, NY) for 30 min at 4°C in PBS, cells were resuspended in Fc-Block (BioLegend) containing PBS/2% FCS. Cells were then simultaneously stained with oligonucleotide-conjugated antibodies targeting CD24, CD44, CD45RB, CD73, SLAMF1, SLAMF6, Vγ1, and Vγ4 (TotalSeqC-BioLegend), oligonucleotide-conjugated hashtag antibodies (TotalSeqC-BioLegend), and fluorophore-conjugated antibodies for cell sorting which included the anti-CD11b (M1/70), anti-CD11c (N418), anti-CD19 (1D3), anti-CD45.2 (104), and anti-TCRβ (H57-597) antibodies. Following surface staining of the enriched population, TCRδ^+ cells were sorted from a CD11b^-CD11c^-CD19^- TCRβ^- population of CD45.2^+ live lymphocytes.

Libraries were prepared according to the manufacturer's instructions with modifications to the V(D) J enrichment procedure. Equal numbers of live GL3^+ γδ T cells from B6 and B6.*Sh2d1a*^-/- mice were pooled and captured (between 5000 and 7000 cells) using a 10x Chromium controller. Following cDNA preparation, the gene expression and CSP libraries were created according to the manufacturer's instructions. To obtain TCR gamma and delta chain sequences, we replaced kit-provided enrichment primers with custom mouse γ and δ chain-specific primers and modified the number of PCR cycles used from manufacturer recommended cycling conditions. In the first enrichment step, an equal amount of cDNA was used as a template for PCR amplification of γ and δ chains separately using a total of 12 and 10 PCR cycles, respectively. The primers used for the first γ-chain enrichment are: GDC1F Forward Primer 5′-AATGAT ACGGCGA CCACCG AG ATCTACACTCTT TCCCTACAC GACGCTC-3′ (2 µM) and GC1R1 Reverse Outer Primers 5′-GGAA AGAACTT TTCAAGGAS ACAAAG-3′ (1 µM), GC1R2 5′-CCCTTATG ACTTCAGG AAAGAA CTTT-3′ (0.5 µM). The primers used for the first δ-chain enrichment are: GDC1F Forward Primer 5′-AATGAT ACGGCGACCACCG AGATCTACACTC TTTC-CCTAC ACGACGCTC-3′ (2 µM) and DC1R1 Reverse Outer Primer 5′-CCACAATC TTCTTGGATG ATCTGAG-3′ (0.5 µM). Following PCR product purification, the γ- and δ-chain amplicons went through the second round of enrichment separately using a total of 12 and 10 PCR cycles, respectively. The primers used for the second γ-chain enrichment are: GDC2F Forward Primer 5′-AATGA TACGGCGA CCA CCGAGATCT-3′ (0.5 µM) and GC2R1 Reverse Inner Primers 5′-ACAAA GGTATGTCCCA GTCT TATGGA-3′ (0.5 µM), GC2R2 5′-GGAGACA AAGGTAGGT CCCAGC-3′ (0.5 µM). The primers used for the second δ-chain enrichment are: GDC2F Forward Primer 5′-AATGATA CGGCGACCACC GAGATCT-3′ (0.5 µM) and DC2R1 Reverse Inner Primer 5′-GTCACCTC TTTAGGGTAG AAATCTT-3′. Following PCR product cleanup, the second round PCR products (γ- and δ-chain amplicons) were quantified and combined in equal quantity making the final TCR V(D)J library. Finally, all three libraries (gene expression, cell surface protein expression, and TCR V(D)J) were sequenced (paired-end 26 × 91 bp) on an Illumina HiSeq 1500 sequencer or a NextSeq 2000.

## CITE-seq quality control, filtering, and demultiplexing

Raw sequence reads from the gene expression and CSP libraries were processed using the Cell Ranger software (v4.0.0) (*Zheng et al., 2017*). Reads were mapped to the GRCm38 (mm10) mouse reference genome. To exclude the possibility of TCR transcripts playing a role in UMAP clustering, all TCR genes were omitted from the analysis using the 'cellranger reanalyze' argument of Cell Ranger software.

Quality control, filtering, UMAP projection, and clustering analysis were conducted using Seurat (v4.3.0.1) (*Hao et al., 2021*) R package. Briefly, cells with a low number (≤2000 for E17 thymus dataset, ≤1200 for D9 thymus dataset and ≤750 for lung dataset) of unique genes (nFeature_RNA), a frequency of mitochondrial reads >4% and a frequency of ribosomal protein <~12% were eliminated from further analysis. To reduce cell cycle- and stress-related heterogeneity in the normalized data, selected

cell cycle genes and stress-induced genes (*Denisenko et al., 2020*) were regressed out during the data normalization step. Note that in some instances, contaminating clusters expressing macrophage specific marker genes were manually removed using Seurat 'CellSelector ()' argument. We used the Seurat function 'HTODemux()' to demultiplex single cells back to their sample of origin (*Stoeckius et al., 2018*).

## CITE-Seq and V(D)J enrichment data analysis

The demultiplexed datasets were further analyzed for gene expression, cell surface protein expression following the multimodal analysis pipeline of Seurat (*Hao et al., 2021*). For each dataset, we performed normalization and variance stabilization of the gene expression data using the scTransform (*Hafemeister and Satija, 2019*). Following identification of top significant principal components (PCs) using the principal component analysis, we performed a graph-based (UMAP) semi-supervised clustering of cells with similar gene expression patterns using specific number of top PCs (E17 = 25, D9 = 30, W6 = 27, lung = 28) and resolution values (E17 = 1.2, D9 = 1.6, W6 = 0.65, lung = 1.2) for each dataset. The cell surface protein expression data (Centered Log Ratio normalized) from CSP libraries as well as the gene expression data were then visualized on the clustered cells using the R package scCustomize (*Marsh, 2021*). The differentially expressed features of each cluster (cluster biomarkers) were identified using FindAllMarkers() function using the Wilcoxon Rank Sum test. For identifying differentially expressed genes between B6 and B6.*Sh2d1a*$^{-/-}$ samples, we performed a pseudobulk DE (differential expression) analysis of aggregated read counts using the DEseq2 package, which accounts for biological replicates making it a better approach than other specialized single-cell DE analysis methods (*Squair et al., 2021*). Briefly, for cells of a given sample type, we aggregated RNAseq reads across biological replicates identified by individual hashtags. Then transformed the genes-by-cell matrix to a genes-by-replicates matrix and performed DE analysis using the DESeq2 (*Love et al., 2014*) package employing the Wald test of the negative binomial model coefficients to determine statistical significance. Finally, the trajectory and pseudotime inference analysis was performed using the partition-based graph abstraction (*Wolf et al., 2019*) analysis, which is a part of the Scanpy (*Wolf et al., 2018*) python package.

For the analysis of V(D)J library sequences, Cell Ranger v6.1.2 ('cellranger vdj' argument) was used to map the TCR V(D)J reads to a custom-made reference containing sequences from the IMGT database (*Lefranc, 2011*). The Cell Ranger V(D)J output was then processed with custom bashscripts. Briefly, the TRGV and TRAV/TRDV contigs identified by Cell Ranger V(D)J were demultiplexed to individual barcodes after which the dominant TRGV and TRAV/DV transcripts were selected per cell. Finally, the V(D)J information of each cell was integrated with scCITE-seq data and visualized using a combination of Seurat and custom-made R scripts. We note that since about half of Vγ1, Vγ4, and Vγ7 T cells contain functionally rearranged TRGV2 chains, but only a small percentage express Vγ2 on the cell surface (*Pereira and Boucontet, 2004*), the methods used here cannot adequately assess the true TRGV2 repertoire. Therefore, we report it here for accuracy but have omitted TRGV2 from our analysis.

## Quantitative PCR

Lung qPCR samples were prepared by positively selecting leukocytes using CD45 MicroBeads (Miltenyi Biotec). Spleen qPCR samples were prepared by depletion of CD19+TCRβ+ cells using biotinylated antibodies and Streptavidin MicroBeads (Miltenyi Biotec) according to the manufacturer's instructions. Total RNA was extracted from approximately 5–10 million cells from each organ using the RNeasy Mini Kit (QIAGEN). SuperScript IV Reverse Transcriptase (Thermo Fisher Scientific, Grand Island, NY) was used to generate cDNA using 400 ng of RNA and oligodT primer according to the manufacturer's instructions. qPCR was performed using 1 µl of cDNA template in iTaq Universal SYBR Green Supermix (Bio-Rad) containing TCR delta chain-specific primer pairs. The qPCR reaction was run on a Bio-Rad CFX96 qPCR machine using the following cycling conditions: 95°C for 30 s; 40 cycles of [95°C for 5 s; 60°C for 30 s; 72°C for 30 s]. The β2M and TRDC genes were used as endogenous housekeeping controls. Primers used were as follows: mb2mforward, 5'-CATGGCTCG CTCGGTGA CC-3', mb2mreverse, 5'-AATGTGAGGC GGGTGGAACTG-3', TRDCforward, 5'-CTCCGGCCA AACC ATCTGTT-3', mouseDV2, 5'-CCAAGAAG CATACAAGCAGT ATAATG-3', mouseDV5, 5'-CCCATGA TGCAGATTT TGTTCAAGG-3', mouseDV7, 5'-GGAAG MCTCGTCAGC CTGTTGT-3', mouseDCrev,

5′-CCACAATCTT CTTGGAT GATCTGAG-3′. Some primer sequences were adopted from *Wei et al., 2015*.

## Vγ4 single-cell sorting and TCR sequencing

The plate-based based γδ TCR repertoire analysis of single-cell sorted Vγ4 cells involved adult lungs from 8- to 10-week-old B6 male (*n* = 7) and female mice (*n* = 6). Lung Vγ4 TCR sequences were obtained using a protocol modified from *Wei et al., 2015*. Briefly, lung cell suspensions were prepared as described above. After washing, cells were resuspended in Fc-Block (BioLegend) containing PBS/2% FCS buffer, and stained with anti-CD11b (M1/70), anti-CD11c (N418), anti-CD19 (1D3), anti-CD45 (30-F11), anti-TCRδ (GL3), and anti-Vγ4 (UC3-10A6) from for 30 min at 4°C in PBS. Fixable Viability Dye eFluor 780 (Thermo Fisher Scientific, Grand Island, NY) was used to identify dead cells. Single-cell sorting of lung Vγ4$^+$TCRδ$^+$ from a CD11b$^-$CD11c$^-$CD19$^-$TCRβ$^-$ population of CD45$^+$ live lymphocytes was performed using a FACS Aria III cell sorter (BD Bioscience). Cells were sorted directly into 12.5 µl (per well) of OneStep RT-PCR Buffer mix (QIAGEN) in a 96-well PCR plate (one cell/well).

TCR sequences were then amplified and barcoded according to *Wei et al., 2015*. Briefly, the first two nested PCR reactions involved primer-specific amplification of gamma and delta TCR sequences. In both the first and second PCR reactions, the final concentration of each V-region primer is 0.36 µM, and each C-region primer is 0.6 µM. The first RT-PCR step was performed using the following cycling conditions: 50°C for 30 min; 95°C for 5 min; 25 cycles of [94°C for 30 s; 62°C for 1 min; 72°C for1 min]; 72°C for 7 min; 4°C. Next, 1 µl of the first round PCR product was used as a template for the second nested PCR reaction using HotStartTaq Master Mix Kit (QIAGEN). The cycling conditions for the second PCR reaction were 95°C for 5 min; 25 cycles of [94°C for 30 s; 64°C for 1 min; 72°C for 1 min]; 72°C for 7 min; 4°C. In the third PCR reaction, 1 µl of second round PCR product as the template in a new 14 µl reaction using HotStartTaq Master Mix Kit (QIAGEN). Each well was barcoded. The final concentration of 5′ and 3′ barcoding primers was 0.05 µM each and paired-end primers were 0.5 µM each. The cycling conditions for the third PCR reaction were 95°C for 5 min; 36 cycles of [94°C for 30 s; 62°C for 1 min; 72°C for 1 min]; 72°C for 7 min; 4°C. The γ and δ chain products were run on 1.2% agarose gel separately and purified using the QIAGEN QIAquick Gel Extraction Kit according to kit instructions. The final products were then analyzed using the 2100 Bioanalyzer (Agilent Technologies). Finally, the γ and δ TCR products were mixed in equal concentration and sequenced on the Illumina MiSeq platform using the Illumina MiSeq Reagent Kit v2 (500-cycles).

The paired-end TCR sequencing reads were analyzed using a custom software pipeline capable of de-multiplexing barcoded sequences to individual cells. The resulting de-multiplexed reads were then analyzed by MiGMAP software, which identifies CDR3 sequences using the IgBlast 1.4.0 (NCBI) V(D)J mapping tool (*Shugay et al., 2015*; *Ye et al., 2013*). The single-cell TCR V(D)J profiling data were then visualized using custom R scripts.

## Statistical analysis

Statistical analysis was conducted using Prism (GraphPad Software, San Diego, CA). Unpaired Student *t*-tests, one-way ANOVA, or two-way ANOVA with Sidak multiple comparisons corrections were used where appropriate. Sample size estimates were calculated using StatMate (GraphPad Software, San Diego, CA). Statistical tests were considered significant when p-values were less than 0.05. Error bars represent standard deviation unless otherwise indicated.

## Acknowledgements

We thank Marlana Winschel, Cameron Moquin, and Erin Mathieu for help in tissue preparation and flow cytometry. We acknowledge Roxana del Rio Guerra for help in cell sorting. All flow cytometry data were collected at the Harry Hood Bassett Flow Cytometry and Cell Sorting Facility (Larner College of Medicine, University of Vermont) and was supported by NIH grant (S10OD026843). We thank Jessica Hoffman, Kris Finstaad, and the Vermont Integrated Genome Resource (Larner College of Medicine, University of Vermont) for help in scRNAseq, Julie Dragon and Korin Eckstrom for help in bioinformatics analysis. We thank Fred Kolling at the Genomics Shared Resource (Dartmouth Geisel School of Medicine) for help in sequencing. We also thank Willi Born and Rebecca O'Brien for the 17D1 hybridoma and Pablo Pereira for helpful discussion. This work benefitted from data assembled

by the ImmGen consortium, and was supported by NIH grants R21AIO66465 and R03AI153902, S10OD026843 (JEB) and an AAI Careers in Immunology Fellowship (SKM, JEB).

## Additional information

### Funding

| Funder | Grant reference number | Author |
|---|---|---|
| National Institutes of Health | R21AI166465 | Jonathan E Boyson |
| National Institutes of Health | R03AI153902 | Jonathan E Boyson |
| American Association of Immunologists | | Somen K Mistri |
| National Institutes of Health | S10OD026843 | Jonathan E Boyson |

The funders had no role in study design, data collection, and interpretation, or the decision to submit the work for publication.

### Author contributions

Somen K Mistri, Conceptualization, Formal analysis, Investigation, Methodology, Writing – original draft, Writing – review and editing; Brianna M Hilton, Katherine J Horrigan, Emma S Andretta, Formal analysis, Investigation; Remi Savard, Oliver Dienz, Investigation; Kenneth J Hampel, Resources, Formal analysis, Investigation; Diana L Gerrard, Resources, Investigation, Methodology; Joshua T Rose, Validation, Investigation, Methodology; Nikoletta Sidiropoulos, Supervision; Dev Majumdar, Resources, Formal analysis, Validation, Methodology; Jonathan E Boyson, Conceptualization, Data curation, Formal analysis, Supervision, Funding acquisition, Investigation, Methodology, Writing – original draft, Project administration, Writing – review and editing

### Author ORCIDs

Somen K Mistri ⓘ http://orcid.org/0000-0002-2139-6927
Brianna M Hilton ⓘ http://orcid.org/0009-0001-5209-7268
Katherine J Horrigan ⓘ http://orcid.org/0009-0001-3102-3315
Oliver Dienz ⓘ http://orcid.org/0000-0001-9380-4873
Jonathan E Boyson ⓘ https://orcid.org/0000-0003-2673-9148

### Ethics

All experimental procedures on animals were carried out with the approval of the University of Vermont Institutional Animal Care and Use Committee. Protocols PROTO202000073 and PROTO202000179.

Reviewer #1 (Public review): https://doi.org/10.7554/eLife.97229.3.sa1
Reviewer #2 (Public review): https://doi.org/10.7554/eLife.97229.3.sa2
Author response https://doi.org/10.7554/eLife.97229.3.sa3

## Additional files

### Supplementary files

• Supplementary file 1. Cluster-specific gene markers for thymic gamma delta T cells.

• Supplementary file 2. TCR clonotypes in B6 and B6.Sh2d1a-/- thymic gamma delta T cells.

• Supplementary file 3. Pseudobulk analysis of differential gene expression among immature B6 and B6.Sh2d1a-/- thymic gamma delta T cells.

• Supplementary file 4. Bulk RNAseq differential gene expression between neonatal immature B6 and B6.Sh2d1a-/- gamma delta T cells.

• Supplementary file 5. Cluster-specific gene markers for adult lung gamma delta T cells.

• Supplementary file 6. TCR clonotypes in B6 and B6.Sh2d1a-/- lung gamma delta T cells.
• MDAR checklist

## Data availability

All sequencing data in this manuscript have been deposited in NCBI Gene Expression Omnibus (GEO) and have been assigned the accession number GSE262064. All scripts used in these analyses have been made publicly available on Github and can be accessed here: https://github.com/Boyson-Lab/scRNAseq_with_immune_profiling_tutorial (copy archived at *Mistri, 2024*). All data generated or analyzed during this study are included in the manuscript and supporting files; source data files have been provided.

The following dataset was generated:

| Author(s) | Year | Dataset title | Dataset URL | Database and Identifier |
|---|---|---|---|---|
| Mistri SK, Hilton BM, Horrigan KJ, Andretta ES, Savard R, Dienz O, Hampel KJ, Gerrard DL, Rose JT, Sidiropoulos N, Majumdar D, Boyson JE | 2024 | SLAM/SAP signaling regulates discrete γδ T cell developmental checkpoints and shapes the innate-like γδ TCR repertoire | https://www.ncbi.nlm.nih.gov/geo/query/acc.cgi?acc=GSE262064 | NCBI Gene Expression Omnibus, GSE262064 |

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
