## [Editor Report · eLife assessment]

This **important** study highlights the importance of SLAM-SAP signaling in determining innate gamma-delta T cell sublineages and their T cell receptor repertoires. It uncovers the complex role of the SLAM-SAP pathway in developing specific gamma-delta T cell subsets. The evidence presented is **compelling**, backed by high-quality data obtained through advanced single cell proteogenomics techniques.This work will be of broad interest to immunologists.

---

## [Referee Report · Reviewer #1 (Public review)]

Summary:

In this study the authors advance their previous findings on the role of the SLAM-SAP signaling pathway in the development and function of multiple innate-like gamma-delta T cell subsets. Using high throughput single cell proteogenomics approach, the authors uncover SAP-dependent developmental checkpoints, and the role of SAP signaling in regulating the diversion of γδ T cells into the αβ T cell developmental pathway. Finally, the authors define TRGV4/TRAV13-4(DV7)-expressing T cells as a novel, SAP-dependent Vγ4 γδT1 subset.

Strengths:

This study furthers our understanding of the importance and complexity of the SLAM-SAP signaling pathway not only in the development of innate-like γδ T cells but also the how it potentially balances the γδ/αβ T cell lineage commitment. Additionally, this study reveals the role of SAP-dependent events in generation of γδ TCR repertoire.

Comments on revised version:

The conclusions of the study are supported by well thought-out experiments and compelling data.

Weaknesses:

There are no major weakness in the study.

A few minor points:

(1) In the subsets of the γδ T cells that exhibit reduced BLK expression in B6. SAP KO mice, have the authors examined the expression of Lck and/or Fyn?

(2) Does BLK directly associate with SLAM F1 and or SLAM F6 receptors?

(3) Given the emerging role of γδ T cells in host immunity, it will be useful if the authors add a discussion of how their findings are relevant in disease conditions such as in cancer.

The author has adequately addressed all the reviewers' comments.

---

## [Referee Report · Reviewer #2 (Public review)]

Summary:

Mistri et al explore the role of SLAM-SAP signaling in the developmental programming of innate-like gd T cell subsets. Using proteo-genomics, they determined that abrogation of SLAM-SAP signaling altered that programming, reducing some IL-17 producing subsets, including a novel Vγ4 γδT1 subset, and diverting gdTCR-expressing precursors to the ab fate. Altogether, this is a very thorough, thoughtfully interpreted study that adds significantly to our understanding of the contribution of the SLAM-SAP pathway to lineage specification. A particularly interesting element is the role of SLAM-SAP in preventing gd17 progenitors from switching fates and adopting the ab fate.

Comments on revised version:

The authors have addressed the minor issues raised in the original submission.

---

## [Author Response]

The following is the authors’ response to the original reviews.

**Reviewer #1:**
(1) In the subsets of the γδ T cells that exhibit reduced BLK expression in B6. SAP KO mice, have the authors examined the expression of Lck and/or Fyn?

The reviewer raises an excellent point. We have included in the revised manuscript additional data on Lck and Fyn expression in our scRNAseq dataset in (new Suppl. Fig. 1 and new Suppl. Fig. 4). These data revealed that in contrast to Blk, which appears primarily restricted to the γδT17 clusters, Lck and Fyn exhibit a much broader distribution and lack restriction to specific clusters. We did note that, like Blk, Lck and Fyn transcripts were abundant in SAP-dependent C2 cluster cells. Pseudobulk analysis on the immature clusters revealed that, neither Fyn nor Lck expression level differences reached our cut-off of 0.5 log2 FC (log2 FC Blk = 1.06), leading us to conclude that Blk is particularly dependent on SAP. We did note, however, that the magnitude of Lck differential expression was close to the 0.5 log2 FC cut-off and that its expression was increased in B6.SAP-/- γδ T cells (Suppl. Fig. 4). These results have been added to lines 202-212 in the Results section and lines 491-499 in the Discussion section.

(2) Does BLK directly associate with SLAM F1 and or SLAM F6 receptors?

The reviewer raises an interesting question given previous reports that BLK, LCK, and FYN have all been implicated in γδ T cell development. While SAP has a well-known ability to recruit FYN to SLAMF1 and there is evidence of a similar SAP-mediated recruitment of LCK to SLAMF6, we are not aware of any evidence a SAP-BLK interaction or of a direct binding of BLK to SLAM family receptors. Future experiments to investigate this possiibility are certainly warranted. In the revised ms, we have included additional discussion of these possibilities (lines 491- 499).

(3) Given the emerging role of γδ T cells in host immunity, it would be useful if the authors could add a discussion of how their findings are relevant in disease conditions such as cancer.

We agree and have included new text in the Introduction (lines 37-45).

(4) Delete repeated words in lines 546 and line 553.

Thank you—this has been corrected in the revised manuscript.

**Reviewer #2:**
This is a very complete study and requires no additional experimentation. One thing to keep in mind in assessing the ultimate fate of the "ab wannabe cells" is that mechanisms exist to silence the gd TCR as cells differentiate to the DP stage and so their presence as diverted DP cells may not be evident by staining for gdTCR expression - and will only be evident transcriptomically.

We appreciate this helpful comment from the reviewer which we will take into consideration in our future experimental design.

There are a couple of minor points to raise:(1) Figure 3C is not called out in the text.

Thank you—this has been corrected in the revised manuscript.

(2) Line 546 - "dependent" is repeated.

Thank you—this has been corrected in the revised manuscript.